# Evidence for a major missing source in the global chloromethane budget from stable carbon isotopes

Enno Bahlmann[1,2], Frank Keppler[3,4,5], Julian Wittmer[6,7], Markus Greule[3,4], Heinz Friedrich Schöler[3], Richard Seifert[1], Cornelius Zetzsch[5,6]

[1]Institute of Geology, University Hamburg, Bundesstrasse 55, 20146 Hamburg, Germany
[2]Leibniz Centre for Tropical Marine Research, Fahrenheitstraße 6, 28359 Bremen
[3]Institute of Earth Sciences, Heidelberg University, Im Neuenheimer Feld 234-236, 69120 Heidelberg, Germany
[4]Heidelberg Center for the Environment (HCE), Heidelberg University, 69120 Heidelberg, Germany
[5]Max-Planck-Institute for Chemistry, Hahn-Meitner-Weg 1, 55128 Mainz, Germany
[6]Atmospheric Chemistry Research Unit, BayCEER, University of Bayreuth, Dr Hans-Frisch Strasse 1–3, 95448 Bayreuth, Germany
[7]Agilent Technologies Sales & Services GmbH & Co. KG, Hewlett-Packard-Str. 8, 76337 Waldbronn, Germany

*Correspondence to*: Enno Bahlmann enno.bahlmann@leibniz-zmt.de

**Abstract.** Chloromethane ($CH_3Cl$) is the most important natural input of reactive chlorine to the stratosphere, contributing about 16% to stratospheric ozone depletion. Due to the phase out of anthropogenic emissions of chlorofluorocarbons, $CH_3Cl$ will largely control future levels of stratospheric chlorine.

The tropical rainforest is commonly assumed to be the strongest single $CH_3Cl$ source, contributing over half of the global annual emissions of about 4000 to 5000 Gg ($1\ Gg = 10^9\ g$). This source shows a characteristic carbon isotope fingerprint, making isotopic investigations a promising tool for improving its atmospheric budget. Applying carbon isotopes to better constrain the atmospheric budget of $CH_3Cl$ requires sound information on the kinetic isotope effects for the main sink processes: the reaction with OH and Cl in the troposphere. We conducted photochemical $CH_3Cl$ degradation experiments in a 3500 $dm^3$ smog chamber to determine the carbon isotope effect ($\varepsilon = k^{13}C/k^{12}C-1$) for the reaction of $CH_3Cl$ with OH and Cl. For the reaction of $CH_3Cl$ with OH, we determined an $\varepsilon$ value of (-11.2 ± 0.8) ‰ (n=3) and for the reaction with Cl we found an $\varepsilon$ value of (-10.2 ± 0.5) ‰ (n=1) being five to six times smaller than previously reported. Our smaller isotope effects are strongly supported by the lack of any significant seasonal covariation in previously reported tropospheric $\delta^{13}C(CH_3Cl)$ values with the OH driven seasonal cycle in tropospheric mixing ratios.

Applying these new values for the carbon isotope effect to the global $CH_3Cl$ budget using a simple two hemispheric box model, we derive a tropical rainforest $CH_3Cl$ source of (670 ± 200) Gg $a^{-1}$, which is considerably smaller than previous estimates. A revision of previous bottom up estimates, using above ground biomass instead of rainforest area, strongly supports this lower estimate. Finally, our results suggest a large unknown $CH_3Cl$ source of (1530 ± 200) Gg $a^{-1}$.

## 1    1 Introduction

In the mid-90s, the recognition that the known $CH_3Cl$ sources, mainly biomass burning and marine emissions, are insufficient to balance the known atmospheric sinks (Butler, 2000) motivated intense research on potential terrestrial sources. Today, it is common thinking that large emissions from tropical rainforests (Monzka et al., 2010; Xiao et al, 2010; Carpenter et al., 2014) can close this gap. Several model studies revealed a strong tropical $CH_3Cl$ source in the

range of 2000 Gg a$^{-1}$ (Xiao et al., 2010; Yoshida et al., 2006; Lee Taylor et al., 1998). Particular support for a strong tropical rainforest source arose from observations of elevated CH$_3$Cl concentrations in the vicinity of tropical rainforests (Yokouchi et al., 2000), greenhouse experiments (Yokouchi et al., 2002), several field measurements in tropical rainforests (Saito et al., 2008; Gebhardt et al., 2008; Blei et al., 2010; Saito et al., 2013) and from carbon

stable isotope mass balances (Keppler et al., 2005; Saito & Yokouchi, 2008). The majority of CH$_3$Cl in tropical rain forests ((2000 ± 600) Gg a$^{-1}$) is thought to originate from higher plants (Monzka et al., 2010; Xiao et al, 2010; Yokouchi et al. 2000; Saito & Yokouchi, 2008). A minor fraction of about 150 Gg a$^{-1}$ may be emitted from wood rotting fungi (Monzka et al., 2010; Xiao et al, 2010; Carpenter et al., 2014). Further emissions from senescent leaf litter (Keppler et al., 2005) may substantially contribute to this source, but this has not yet been confirmed in field

studies (Blei et al., 2010). On a global scale, biomass burning (400 to 1100 Gg a$^{-1}$) and surface ocean net emissions (140 to 640 Gg a$^{-1}$) are further important sources (Monzka et al., 2010; Xiao et al, 2010; Carpenter et al., 2014). Chloromethane from higher plants has an average stable isotope signature ($^{13}$C/$^{12}$C ratio, δ$^{13}$C value) of (-83 ± 15) ‰ (Saito et al., 2008; Saito & Yokouchi, 2008). Compared to the other known sources with δ$^{13}$C values in the range from -36 ‰ to -62 ‰ (Keppler et al., 2005; Saito & Yokouchi, 2008), the tropical rainforest source is exceptionally depleted

in $^{13}$C making stable isotope approaches particular useful to better constrain CH$_3$Cl flux estimates.

The isotopic composition of tropospheric CH$_3$Cl links the isotopic source signatures to the kinetic isotope effects (KIEs) of the sinks. The primary CH$_3$Cl sink is its oxidation in the troposphere by OH and Cl, accounting for about 80% of total losses (Monzka et al., 2010; Xiao et al, 2010; Carpenter et al., 2014). Further sinks comprise soil uptake and loss to the stratosphere (Monzka et al., 2010; Xiao et al, 2010; Carpenter et al., 2014). An accurate determination

of the KIEs of the main tropospheric sink reactions (CH$_3$Cl + OH, CH$_3$Cl + Cl) is crucial for constraining the tropical rainforest source from an isotopic perspective. A previous study (Gola et al., 2005) revealed large ε values of (-59 ± 8) ‰ and (-70 ± 10) ‰ for the reaction of CH$_3$Cl with OH and Cl, respectively, which supported the hypothesis of large emissions from tropical rainforests (Keppler et al., 2005; Saito & Yokouchi, 2008). In particular, the ε value for the reaction with OH is much larger in comparison to previously reported ε values for the reaction of OH with methane

(Saueressig et al., 2001) and other hydrocarbons (Rudolph et al., 2000; Anderson et al., 2004). We thus performed photochemical degradation experiments of CH$_3$Cl in a 3500 dm$^3$ Teflon smog chamber using established radical generation schemes (see method section for details) to reassess the KIEs for the reaction of CH$_3$Cl with OH and Cl. For validation purposes, we further determined the known KIEs for the same reactions of methane (CH$_4$).

In the next step, we used the seasonal variations in the mixing ratios (Prinn et al., 2000) and isotopic composition

(Thompson et al., 2002; Redeker et al., 2007) of tropospheric CH$_3$Cl to further assess the reliability of the obtained KIEs. This was done with a simple two-box model, dividing the atmosphere into a northern and a southern hemisphere and using a simplified emission scheme. The same model was then used to constrain the tropical rainforest source from an isotopic perspective. We finally improved previous bottom up estimates of the tropical rainforest source using carbon density maps of the tropical rainforest instead of coverage area.

## 2   The kinetic isotope effect (ε) for the reaction of CH₃Cl with OH and Cl

### 2.1   Materials and methods

#### 2.1.1   Smog chamber

The smog chamber setup and the experimental conditions are the same as recently described in Keppler et al. (2018). The samples for the carbon isotope analysis were taken from the same experiments described therein. Briefly, the isotope fractionation experiments were performed in a $(3500 \pm 100)$ dm³ Teflon smog-chamber. The chamber was continuously flushed with purified, hydrocarbon-free zero air (zero-air-generator, cmc instruments, <1 nmol mol⁻¹ of $O_3$, <500 pmol mol⁻¹ $NO_X$, <100 nmol mol⁻¹ of $CH_4$) at a rate of 0.6 - 4 dm³ min⁻¹ to maintain a slight overpressure of 0.5-1 Pa logged with a differential pressure sensor (Kalinsky Elektronik DS1). A Teflon fan inside the chamber ensured constant mixing throughout the experiments. NO and $NO_X$ were monitored on a routine basis with an EcoPhysics, CLD 88p chemiluminiscence analyzer coupled with an EcoPhysics photolytic converter, PLC 860. Ozone was monitored by a chemiluminescence analyzer (UPK 8001). Initial CH₃Cl mixing ratios were between 5 µmol mol⁻¹ and 14 µmol mol⁻¹. Perfluorohexane (PFH) was used as an internal standard with initial mixing ratios of $(25\pm3)$ µmol mol⁻¹ to correct the resulting concentrations for dilution. The mixing ratios of CH₃Cl and PFH were monitored by GC-MS (Agilent Technologies, Palo Alto, CA) with a time resolution of 15 minutes throughout the experiments. The stability of the instrument was regularly checked using a gaseous standard (5 ml of 100 µmol mol⁻¹ CH₃Cl in $N_2$). Mixing ratios of methane and $CO_2$, used as internal standard in the methane degradation experiments, were measured with a Picarro G221i cavity ring down spectrometer. Prior to the experiments, the instrument was calibrated with pressurized ambient air from a tank obtained from the Max-Planck-Institute for Biogeochemistry in Jena/Germany ($CO_2$ mixing ratio of $(394.6\pm0.5)$ µmol mol⁻¹, methane mixing of $(1.752\pm0.002)$ µmol mol-1). OH radicals were generated via the photolysis of ozone (about 2000 nmol mol⁻¹ for CH₃Cl and about 10000 nmol mol⁻¹ for $CH_4$) at 253.7 nm in the presence of water vapour (Relative humidity = 70 %). This is a well-established efficient method for OH radical generation (Cantrell et al., 1990; DeMore, 1992). In the CH₃Cl + OH experiments, initially 2000 µmol mol⁻¹ of $H_2$ was added for scavenging chlorine radicals originating from the photolysis or oxidation of formyl chloride (HCOCl) occurring as an intermediate in the reaction cascade (Gola et al., 2005). To obtain an efficient OH formation, Philips TUV lamps (1x55 W for CH₃Cl, 4x55 W for $CH_4$) were welded in Teflon film and mounted inside or around the smog chamber. Atomic chlorine (Cl) was generated via photolysis of molecular chlorine ($Cl_2$) at a relative humidity of less than 1 % by a solar simulator (Behnke et al., 1988) with an actinic flux comparable to the sun in mid-summer in Germany and a photolysis frequency of $J = 1.55 \times 10^{-3}$ s⁻¹ for $Cl_2$ (Buxmann et al., 2012). A more detailed description of the smog chamber setup is provided in the supplementary information and has recently been published elsewhere (Wittmer et al., 2015; Keppler et al., 2018).

#### 2.1.2   Sampling and carbon isotope determination

From each experiment 10 to 15 canister samples (2 dm³ stainless steel, evacuated <1.3x10⁻³ Pa and baked out at 250°C for 2 h) and 10 to 15 adsorption tube  samples were taken at regular time intervals for subsequent analysis of carbon isotope ratios. The adsorption tubes were made of stainless steel (1/4 inch outer diameter, 7 inch length) and filled with 77 mg Carboxen 1016® , 215 mg Carbopack X 569®, 80 mg Carboxen® 1003 and 9 mg Tenax® TA in order of the sampling flow direction. The adsorption tube samples and one set of canister samples from the CH₃Cl

degradation experiments were analyzed by 2D-GC-IRMS/MS at the University of Hamburg using the method of Bahlmann et al. (2011). This method has been shown to be free of interferences from other compounds. The precision and reproducibility of the $\delta^{13}$C measurements based on standards were ± 0.6‰ (n = 18) on the 1σ level. In order to assure compliance with VPDB scale, a single component standard of $CH_3Cl$ (100 µmol mol$^{-1}$ in nitrogen, Linde Germany) was calibrated against a certified $CO_2$ reference standard (Air Liquide, Germany, (-26.8± 0.2) ‰ and a solid standard (NIST NBS 18, RM 8543) after offline combustion and analysis via a dual inlet (DI). The results from the DI (n=6) were (-37.2 ±0.1) ‰ for $CH_3Cl$. The respective $\delta^{13}$C values from the GC-GC/IRMS, measured against the machine working gas (Air Liquide, Germany, ((-26.8± 0.2) ‰) were (-36.1±0.2) ‰ resulting in an offset (DI – 2D-GC-IRMS) of -1.1 ‰ for $CH_3Cl$.

The canister samples were analyzed at the University of Heidelberg using a cryogenic pre-concentration system coupled to a GC-C-IRMS system, developed for $\delta^2$H measurements of $CH_3Cl$ (Greule et al., 2013). A combustion reactor filled with copper (II) oxide at 850°C was used to analyze $\delta^{13}$C. The precision and reproducibility of these $\delta^{13}$C measurements based on a $CH_3Cl$ working standard were ± 0.47 ‰ (n = 47) on the 1σ level. The sampled $CH_3Cl$ amounts varied between 0.8 and 15 nmol. Both methods were linear over the whole range of sampled $CH_3Cl$ amounts.

The $\delta^{13}$C values measured in both laboratories generally agreed within ± 1.3 ‰ on the 1σ level. This range is somewhat larger than expected from error propagation and may result from small additional errors of scale adding to the uncertainty. For the purpose of this study no attempts were made to adjust the measured $\delta^{13}$C values. $CH_4$ carbon isotope ratios were only analyzed at the University of Heidelberg.

### 2.1.3    Calculation of ε

The carbon isotope ratios are reported in the δ-notation relative to the VPDB scale (Vienna Pee Dee Belemnite), and the kinetic isotope effect (KIE, symbol ε) is reported in ‰. We applied an orthogonal regression model (Danzer et al., 1995) to derive the kinetic isotope effect for each experiment from the slope of the Rayleigh plot:

$$\varepsilon * \ln(f_t) = ln\left(\frac{\delta^{13}C_t+1}{\delta^{13}C_0+1}\right) \tag{1}$$

with ε being the kinetic isotope effect, $f_t$ being the residual $CH_3Cl$ fraction at time t, $\delta^{13}C_0$ being the initial carbon isotope ratio of the substrate ‰, $\delta^{13}C_t$ being the carbon isotope ratio of the substrate ‰ at time t. To account for the dilution from the air flow through the chamber, the residual fraction ($f_t$) has been calculated from the mixing ratios of $CH_3Cl$ and the inert tracer, PFH, as follows:

$$f_t = \frac{[CH_3Cl_t]*[PFH_0]}{[CH_3Cl_0]*[PFH_t]} \tag{2}$$

Here, [$CH_3Cl$] and [PFH] denote the respective concentrations, and the indices t and 0 refer to time t and zero, respectively. The uncertainty for $f_t$ ranged from 1.4 to 1.8 % on the 1σ level.

### 2.2    Results of the $CH_3Cl$ degradation experiments

In total, we performed six degradation experiments and two control experiments within this study. To perform the degradation experiments within a day, the experimental conditions were modified as indicated in table 1. For the OH experiments in the presence of $CH_4$, the light intensity was increased from 55 W to 220 W, and the steady state ozone mixing ratios were increased from about 620 nmol mol$^{-1}$ to about 3570 nmol mol$^{-1}$. Under these experimental conditions, typically 70 to 80 % of the initial $CH_3Cl$ and $CH_4$ were degraded within 6 to 10h. A more detailed

discussion of the experimental conditions with respect to the OH yields and degradation rates is provided in the supplementary information. Further, the reader is referred to Keppler et al. (2018) reporting on the hydrogen isotope effects from these experiments.

Prior to each degradation experiment, we monitored the ratio of $CH_3Cl$ and perfluorohexane (PFH) for at least 2h to assess potential side reactions and unwanted losses of $CH_3Cl$. For the experiment with chlorine, this was done under dark conditions in the presence of 10 µmol mol$^{-1}$ $Cl_2$. For the OH experiments, this was either done in the absence of light or ozone. None of these tests revealed indication for a measurable loss (1.4 to 2.1 %) of $CH_3Cl$ and thus for any biasing side effects or reactions. In the $CH_4$ degradation experiments, $CO_2$ was used as an internal standard to correct the $CH_4$ mixing ratios for dilution. A control experiment over 9 h, carried out with a dilution flow of 4 dm$^3$ min$^{-1}$ of zero air, revealed a slope ($-0.00118\pm0.00001$) min$^{-1}$ for $CH_4$ loss and a slope of ($-0.00117\pm0.000007$) min$^{-1}$ for the $CO_2$ loss respectively. This corresponds to a dilution flow of ($4.1\pm0.1$) dm$^3$ min$^{-1}$, being in good agreement with the pre-set dilution flow (the major uncertainty in this calculation is the exact volume of the chamber). During this control experiment, the dilution corrected mixing ratio of $CH_4$ changed by less than 0.2 %.

In our study, photolysis of ozone (620 nmol mol$^{-1}$ steady state mixing ratio) in the absence of water vapor (relative humidity <1%) but with 2000 µmol mol$^{-1}$ $H_2$ (experiment 3) resulted in a $CH_3Cl$ degradation of less than 3 % over 10 hours and no measurable change in the isotopic composition of $CH_3Cl$ (-46.8 ‰ at the beginning and -46.1 ‰ after 10h) because of the insufficient OH yield. The reaction rate constants of $O(^1D)$ with $H_2$ and $H_2O$ at 298 K are 1.1 x 10$^{-10}$ and 2.2 x 10$^{-10}$ cm$^3$ s$^{-1}$, respectively (Burkholder et al., 2015). At a relative humidity of 70% (corresponding to 16000 µmol mol$^{-1}$), the reaction with $H_2O$ is by far the main pathway to form OH (with the $H_2$ pathway contributing less than 4% to the OH yield). This is consistent with the previous study, where ozone levels of 300 µmol mol$^{-1}$ were required for a sufficient OH production from $H_2$ (Gola et al., 2005; Sellevåg et al., 2006). For this experiment, the partial lifetime of $CH_3Cl$ with respect to OH can be estimated to be about 330 h. Potential side reactions with $O(^1D)$ were not explicitly investigated in our study but because of the reduced OH yield, this experiment allows to constrain potential losses of $CH_3Cl$ due to reaction with ($O(^1D)$). In experiment four, where both $CH_3Cl$ and $CH_4$ were present, the ratio of the measured rate constants for the reaction of $CH_3Cl$ and $CH_4$ with OH was 5.8. This ratio agrees well with that of the recommended rate constants of 5.6 at 298 K ($6.3x10^{-15}$ cm$^3$ s$^{-1}$ for $CH_4$ and, $3.5x10^{-14}$ cm³ s$^{-1}$ for $CH_3Cl$ at 298K; Burkholder et al., 2015).

The change in stable carbon isotope δ values of $CH_3Cl$ ($\delta^{13}C(CH_3Cl)$) with extent of reaction and the corresponding Rayleigh plots of the $CH_3Cl$ degradation experiments are shown in figure 1. The respective ε values, derived from the slope of the Rayleigh plot, are summarized in table 2. For the reaction of $CH_3Cl$ with OH, we determined an ε value of ($-11.2\pm0.8$) ‰ (n=3) and for the reaction with Cl we found an ε value of ($-10.2\pm0.5$) ‰ (n=1). The results from both laboratories generally agreed within $\pm1.5$ ‰ (1σ) and showed no systematic difference. Variations in the initial mixing ratios (5 to 13 µmol mol$^{-1}$) and isotopic composition (($-47.0\pm0.5$) ‰ and ($-40.3\pm0.5$) ‰) of $CH_3Cl$ in the OH experiments had no significant effect on the determination of the isotope effects. Furthermore, the increase in the light intensity and ozone mixing ratios in experiment four had no effect on the isotope effects.

The ε values for the reaction of $CH_4$ with OH and Cl, determined for validation purposes, agreed reasonable well with the previously published KIEs (Saueressig et al., 1995; 2001; Tyler et al., 2000; Feilberg et al., 2005). For the reaction of $CH_4$ with OH, we found an ε value of -4.7 ‰, being at the upper end in terms of absolute magnitude of previous reported fractionation factors, and for the reaction with Cl we found an ε value of -59‰, being more at the lower end

of previously measured KIEs (table 3). Prior to the $CH_4$ degradation experiment with OH, we performed a control experiment (exp. 5 in table 1) that revealed no $CH_4$ loss over 10 h. With this, we can exclude any interferences from reactive chlorine during the $CH_4$-OH experiment. The larger isotope effect for the reaction with OH found here might result from the reaction of $CH_4$ with $O(^1D)$. Cantrell et al. (1990), who also used UV-photolysis in the presence of

water as an OH source, reported an even higher ε value of (-5.4 ±0.9) ‰ and estimated that the reaction of $CH_4$ with $O(^1D)$ (showing an ε value of -13 ‰; Saueressig et al., 2001) may have contributed about 3% to the overall degradation. Saueressig et al. (2001) reported an ε value of -3.9 ‰ for the reaction of $CH_4$ with OH. With respect to this value, a contribution of 9% from the reaction with $O(^1D)$ is required to explain the difference in terms of $O(^1D)$ loss.

**2.3    Discussion of the $CH_3Cl$ degradation experiments**

Our newly determined isotope effects for the reaction of $CH_3Cl$ with OH and Cl are five to six times smaller than the previous reported ε values of (-59 ±10) ‰ for the reaction with OH and of (-70 ±10) ‰ for the reaction with Cl (Gola et al., 2005). In this section, we first discuss potential sources of error in our study with particular respect to the differences between our study and the Gola study and then provide a more comprehensive comparison of our data

with previous data. Gola et al. (2005) used a 250 $dm^3$ electro polished stainless steel chamber for their degradation experiments. We used a 3500 $dm^3$ smog chamber, made of FEP foil, for the $CH_3Cl$ degradation experiments. The large volume of our smog chamber may result in incomplete mixing and thus in an underestimation of the KIE due to transport limitation. The effect of mixing on the observed KIE can be approximated from the time scales of mixing and reaction according to the following equation (Morgan et al., 2004; Kaiser et al., 2006):

$$\epsilon_{obs} \approx \frac{1}{2}\varepsilon_i \times \left(1 + \sqrt{\frac{1}{1+Q}}\right) \tag{3}$$

Here $\varepsilon_i$ is the intrinsic fractionation factor, $\epsilon_{obs}$ is the observed fractionation factor and Q is the ratio of the mixing time and reaction time scale (1/k). The chemical lifetime of $CH_3Cl$ under the experimental conditions was in the order of 6 to 8 h and the turnover of air inside the chamber occurred on time scales of a few minutes. With a reaction time scale of 300 minutes and a mixing time scale of 10 minutes we obtain ε $_{obs}$ ≈ 0.99 x ε$_i$, making incomplete mixing an

unlikely source of error. Incomplete mixing would also have affected the determination of the respective KIEs for $CH_4$. With those values being within previously reported ε values, we can exclude incomplete mixing as a potential source of error in our experiments. In the Gola et al. (2005) study, the mixing ratios and isotope ratios were determined with long-path FTIR. In our study, the mixing ratios were determined by GC-MS and the isotope ratios were measured by GC-IRMS in two different laboratories. Both labs used different analytical setups, different sampling methods and

different standards. However, the results from both labs generally agree within ±1.5‰ on the 1σ level and show no systematic difference. As outlined before, different initial $\delta^{13}C(CH_3Cl)$ as well as different initial $CH_3Cl$ mixing ratios had no significant effect on the determination of the isotope effects. This makes analytical artefacts in our $\delta^{13}C$ determination unlikely. The Cl radical generation scheme was quite similar among both studies. Gola et al. (2005) used narrow band photolysis of $Cl_2$ employing a Philips TLD-08 fluorescent lamp (λmax ~370) nm whereas we used

broadband photolysis (300 to 700 nm), making this an unlikely source for the discrepancy in between the isotope effects for the reaction of $CH_3Cl$ with Cl.

In our study, OH was generated via UV photolysis of ozone (steady state mixing ratios of 0.62 and 3.6 $\mu$mol mol$^{-1}$) in the presence of water vapor (RH of 70%) and 2000 $\mu$mol mol$^{-1}$ H$_2$, whereas in the Gola study OH was generated in the absence of water vapor from the reaction of O($^1$D) with H$_2$ (2000 $\mu$mol mol$^{-1}$) after UV-photolysis of ozone (300 $\mu$mol mol$^{-1}$). Due to the much lower ozone mixing ratios employed in our study, the OH generation in the absence of

5 water vapor was not sufficient in our study. Both OH generation schemes are well established. However, Cantrell et al. (1990), who used UV-photolysis in the presence of water as an OH source, estimated that the reaction of CH$_4$ with O($^1$D) may contribute about 3% to the overall degradation. The higher ozone levels and the less efficient conversion of O($^1$D) to OH in the Gola et al. (2005) study suggest an overall higher transient O($^1$D) concentration as compared to our experiments. Anyhow, interferences from the reaction with O($^1$D) are less likely for CH$_3$Cl than for CH$_4$. The

10 reaction rate for CH$_4$ with O($^1$D) (1.7 x 10$^{-10}$ cm³ s$^{-1}$; Burkholder et al., 2015) is 2.7x10$^4$ times larger than the respective reaction rate for OH (6.3 x 10$^{-15}$ cm³ s$^{-1}$). In the case of CH$_3$Cl, the ratio is only 7.4 x 10$^3$ (2.6 x 10$^{-10}$ cm³ s$^{-1}$ and 3.5 x 10$^{-14}$ cm³ s$^{-1}$; Burkholder et al., 2015). Assuming a contribution of 9 % from the reaction with O$^1$D in the CH$_4$ experiment, the reaction with O($^1$D) should contribute less than 2.3 % to the observed CH$_3$Cl loss. In the CH$_3$Cl control experiment, all experimental parameters beside the relative humidity and hence the OH yields were comparable to

15 CH$_3$Cl degradation experiments with OH. The CH$_3$Cl loss of less than 3% over 10h can most likely be attributed to reaction with OH, originating from the reaction of O($^1$D) with H$_2$. The $\delta^{13}$C values of CH$_3$Cl were -46.8 ‰ at the beginning and -46.1 ‰ after 10 h, being indistinguishable within the measurement uncertainty. This experiment makes any biasing side reactions unlikely. In any case we can limit the loss from potential side reactions to less than 3 %. In addition, none of our tests prior each degradation experiment revealed indication for a measurable loss of CH$_3$Cl.

With this, we can safely exclude any measurable effect from potential side reactions on the determination of the KIEs in our study.

A comparison of our data with previously measured and calculated isotope effects for the reaction of CH$_3$Cl, CH$_4$ and other VOCs with OH and Cl is provided in table 3. In a follow up study to Gola et al. (2005), Sellevåg et al. (2006) attributed these exceptionally large fractionation factors to higher internal barriers of rotation of the OH radical

compared to the CH$_4$ + OH reaction. Using variational transition state theory, the authors calculated $\varepsilon$ values of -47 ‰ and -37 ‰ for the reaction of CH$_3$Cl with OH and Cl, respectively. However, a simultaneous theoretical study provided an $\varepsilon$ value of only -3.6 ‰ for the reaction of CH$_3$Cl with OH (Jalili, & Akhavan, 2006). For C-H bond breakage, Streitwieser's semi-classical limit for isotope effects is -21 ‰ (Elsner et al., 2005), and for reactions involving hydrogen radical transfer, an $\varepsilon$ value of -15‰, has been reported (Merrigan et al., 1990). Both values support

a lower fractionation factor. For the reaction of ethane with OH, that can be approximately regarded as a substituted methane, an $\varepsilon$ value of (-7.5 $\pm$ 0.5) ‰ has been reported (Piansavan et al., 2017). One can estimate an upper limit for the reactive site by multiplying $\delta^{13}$C with the number of carbon atoms in the molecule (Anderson et al., 2004). This leads to an upper limit of (-15.0 $\pm$ 0.7) ‰ for the $\varepsilon$ value at the reactive center. In line with this, Anderson et al. (2004) reported a group kinetic isotope effect of (-18.7 $\pm$ 5.2) ‰ for the reaction of primary carbon atoms of alkanes with

OH. The same group (Anderson et al., 2007) found a group kinetic isotope effect of (-18.6 $\pm$ 0.3) ‰ for the respective reaction with Cl. Our smaller kinetic isotope effect for the reaction of CH$_3$Cl with OH and Cl are much closer to these group specific kinetic isotope effects than the previously reported ones of Gola et al. (2005).

To this end, the large discrepancy between our data and those of Gola et al. (2005) remains unresolved and cannot be explained from experimental details. However it appears that the authors have not tested the accuracy of their isotope

ratio measurements as a function of the isotopologue mole fraction in the presence of other species with overlapping spectra (HCl, H$_2$O, O$_3$, etc.), e.g. by using a dilution series.

In any case, the strongest support for our smaller isotope effects arises from the absence of any significant seasonal variation in the tropospheric $\delta^{13}$C(CH$_3$Cl), as shown in section 3.5.

## 3    Carbon isotope modelling

### 3.1    Model set up

The model used in this study is similar to previous two box models (Tans, 1997; Sapart et al., 2012; Saltzman et al., 2004; Trudinger et al., 2004). The atmosphere is divided in two well-mixed semi hemispheric boxes, representing the northern and the southern hemisphere, and the interhemispheric exchange time is 360 days. The model simulates the major sources and sinks for both, the lighter ($^{12}$CH$_3$Cl) and the heavier isotopologue ($^{13}$CH$_3$Cl), as described by Sapart et al. (2012) for CH$_4$. The source and sink terms from the Xiao et al. model study (2010) serve as a starting point for our model. We use a simplified mass balance with four source categories having distinct isotopic source signatures: higher plants / unknown, oceans, biomass burning, and other known sources (section 3.3 and table 4). Total net emissions were fixed at 4010 Gg a$^{-1}$ with 2210 Gg a$^{-1}$ in the northern hemisphere and 1800 Gg a$^{-1}$ in the southern hemisphere. For each source category, the carbon isotope source signature was randomly varied within the given uncertainties (table 4). Losses are specified by pseudo first order rate coefficients. The sinks implemented in the model (section 3.4 and table 4) are losses due to the reaction with OH, losses to the surface ocean, losses to soils, and losses to the stratosphere. Seasonal variations were modeled with a time step of 1 day, using monthly averaged source terms. Variations in the source composition were modeled with a time step of 90 days, using annually averaged source terms.

### 3.2    Mixing ratios and isotopic composition of tropospheric CH$_3$Cl

Tropospheric CH$_3$Cl has a mean global mixing ratio of about 540 pmol mol$^{-1}$ (Monzka et al., 2010; Carpenter et al, 2014) and shows a pronounced seasonal cycle with an amplitude of 85 pmol mol$^{-1}$ at northern hemispheric mid-latitudes, (Prinn et al., 2000; Yoshida et al., 2006), reflecting the seasonality in the OH sink (Fig. 2., upper panel). Reported mean $\delta^{13}$C values of tropospheric CH$_3$Cl range from (-36.2 ± 1.9) to (-40.8 ± 3.0) ‰ (Tsunogai et al., 1999; Thompson et al., 2002; Redeker et al., 2007; Bahlmann et al., 2011; Weinberg et al., 2014) suggesting an overall mean $\delta^{13}$C(CH$_3$Cl) of (-37.1± 2.7) ‰. In a year round study carried out in Alert, Canada, Thompson et al. (2002) found no seasonal trend in the tropospheric $\delta^{13}$C(CH$_3$Cl) and no clear correlation between the CH$_3$Cl mixing and isotope ratios (Fig. 2., lower panel). From their data the authors estimated the ε value for the OH sink being less than -5 ‰. Including samples from Frazer Point, Canada, Vancouver, Canada, Houston, Texas and Barring Head, New Zealand this study further revealed no indication for a latitudinal trend in tropospheric $\delta^{13}$C(CH$_3$Cl). The lack of a significant co-variation between the mixing ratios and carbon isotope ratios was confirmed in second year round study (Redeker et al 2007).

### 3.3    Sources

The seasonal source terms are specified for each hemisphere using monthly means as depicted in figure 3. The ocean is treated as a net source for CH$_3$Cl with annual net emission of 335 Gg a$^{-1}$ (range: 80 to 610 Gg a$^{-1}$ (Hu et al., 2013). To account for the bidirectional nature of the gas exchange across the air/sea interface, net fluxes are broken down

into unidirectional gross uptake and emission fluxes, with the uptake carrying the isotopic composition of the atmosphere and the emission carrying the isotopic information of the concurrent formation and degradation processes in the ocean. The gross uptake is calculated using an average transfer velocity of 17 cm h$^{-1}$ for $CO_2$ (Wanninkhof 2014) and a mean tropospheric mixing ratio of 540 pmol mol$^{-1}$. Gross emissions are then calculated as the difference between net emissions and gross uptake fluxes. The reader should note that this approach differs from that of Hu et al. (2013) and results in larger gross fluxes because gross fluxes are calculated for the entire ocean surface. Keppler et al. (2005) estimated average isotopic composition of dissolved $CH_3Cl$ to (-36±4) ‰. This value refers to Komatsu (2004), who reported a mean $\delta^{13}C$ of -38 ‰ for $CH_3Cl$ in coastally influenced waters off Japan and more enriched $\delta^{13}C$ values in the range of -12 ‰ to -30 ‰ from the open North-East Pacific. We obtained average $\delta^{13}C$ values of -43 ±3 ‰ from a productive lagoon in southern Portugal (Weinberg et al. 2014). Taking the biotic and abiotic degradation of $CH_3Cl$ into account, we estimate the mean isotopic source signature of the ocean source to (-36 ±6) ‰. We applied a source strength of 910 Gg a$^{-1}$ for biomass burning (range from 660 to 1230 Gg a$^{-1}$) with 68%, originating from the northern hemisphere, and the emissions peaking during hemispheric spring (Xiao et al., 2010). $CH_3Cl$ from biomass burning shows a $\delta^{13}C$ of (-47 ± 7) ‰ (Czapiewski et al., 2002; Thompson et al., 2002).

The category "other known sources" comprises fungi wetlands and anthropogenic emissions with a total source strength of 365 Gg a$^{-1}$ (range: 79 to 1016 Gg a$^{-1}$) and an averaged isotopic source signature of (-45.5 ± 5.5) ‰ calculated from the source signatures given by Keppler et al. (2005). The emissions from the other known sources were constant over time with 274 Gg a$^{-1}$ in the northern hemisphere and 91 Gg a$^{-1}$ in the southern hemisphere.

The source category "Higher plants / missing" (900 to 3095 Gg a$^{-1}$) represents mainly emissions from the tropical rain forest (900 to 2650 Gg a$^{-1}$) with minor contributions from saltmarshes (80 to 160 Gg a$^{-1}$), rice paddies (5 Gg a$^{-1}$) and mangroves (~50 Gg a$^{-1}$). These emissions are almost equally distributed between both hemispheres and show a slight seasonal peak during hemispheric summer. In order to evaluate the emissions from higher plants, these emissions were divided into two fractions by introducing a split factor. The first fraction represents "true" emissions from higher plants, having an exceptionally depleted isotopic source signature of (-83 ± 15) ‰ (Saito & Yokouchi, 2008; Saito et al., 2008). The second fraction represents an unknown or missing source. The $\delta^{13}C$ of this source is scaled to match the $\delta^{13}C$ of tropospheric $CH_3Cl$. A more depleted $\delta^{13}C$ for this source would point towards additional contributions from a lighter source, such as senescent leaf litter, whereas a more enriched $\delta^{13}C$ for this source points towards additional contributions from a more enriched source.

Saito et al. (2013) recently reported on the bidirectional exchange of $CH_3Cl$ across the leaves of tropical plants with gross uptake rates being roughly 1/6$^{th}$ of gross emission rates. The authors hypothesized that the gross uptake may be related to endosymbiotic bacteria. This uptake might affect the isotopic composition of $CH_3Cl$ from tropical rainforests. However, because the incubation methods used in this study were the same as that previously used to determine the isotopic composition of $CH_3Cl$ emitted from tropical plants (Saito & Yokouchi, 2008; Saito et al., 2008), we can reasonably assume that any potential isotopic effect of this bidirectional exchange is included in the previously reported carbon isotope ratios.

## 3.4   Sinks

The reaction with OH constitutes the single largest sink for $CH_3Cl$, accounting for approximately 80% of its removal from the troposphere. For this study, we used the OH-concentration fields from Spivakovsky et al. (2000) along with

reaction rate constants of Burkholder et al. (2015) to derive monthly resolved lifetimes for both hemispheres. The monthly loss rates were then forced to reproduce seasonal variations of the mixing ratios at Mace Head in the northern hemisphere and at Cape Grim, Tasmania, in the southern hemisphere (Prinn et al., 2000). This resulted in a total tropospheric sink (OH + Cl) of 3614 Gg a$^{-1}$, being comparable to previous modeling studies (Xiao et al., 2010).

In most global budgets, soils are treated as a small sink for chloromethane of about ~250 Gg a$^{-1}$, though a larger uptake exceeding 1000 Gg a$^{-1}$ has been suggested (Keppler et al., 2005; Carpenter et al., 2014). Based on Xiao et al. (2010), we a priori assumed a soil sink of 250 Gg a$^{-1}$ with northern and southern hemispheric fractions of 180 and 70 Gg a$^{-1}$, respectively, reflecting the interhemispheric distribution of the land masses.

   The microbial degradation of CH$_3$Cl in soils is assigned with a large carbon isotope effect of -47 ‰ (Miller et al.,
2001; 2004). The only study, we are aware of (Redeker et al., 2012), that investigates the isotopic composition of soil derived CH$_3$Cl) reports a $\delta^{13}$C of (-34 ± 14) ‰. The soil uptake of CH$_3$Cl can be regarded as a coupled diffusion reaction process, where CH$_3$Cl is first transported into the soil and then undergoes microbial degradation. The apparent isotope effect of such coupled processes will depend on the isotope effects of both steps and can be estimated from diffusion reaction models (Farquhar et al., 1982):

$\varepsilon_{app} = \varepsilon_d + \frac{(\epsilon_m - \varepsilon_d)*(m_d - m_m)}{m_d}$                                              (4)

with $\varepsilon_d$ and $\varepsilon_m$ being the kinetic isotope effects assigned to microbial degradation (47‰) and diffusion (4‰), respectively, and where $m_d$ is the total mass of CH$_3$Cl that enters the soil via diffusion and $m_m$ represents the net soil sink.

   The gross uptake flux ($m_d$) was estimated using a simple transfer resistance model along with the biomes and
respective active seasons as previously employed by Shorter et al. (1995). We used an overall atmospheric transfer resistance governing the transport to the soils surface (aerodynamic transport resistance, quasi-laminar sublayer resistance and in canopy transfer resistance) of 4 s cm$^{-1}$, regardless of the biome, that was derived from reported typical transfer resistances for different biomes (Zhang et al., 2003). The soil uptake is governed by molecular diffusion through the air filled pore space. The soil side transfer resistance can be estimated from the effective diffusion in the
soil column. For a first rough estimate of the soil transfer resistance, we assume an air filled pore space of 0.3 (V/V) and a microbial inactive soil layer of 0.5 cm at the soil surface. Using the Penman model (Penman, 1940) and a diffusion coefficient of 0.144 cm²s$^{-1}$ in air, we obtain a soil transfer resistance of 17 s cm$^{-1}$. With a globally averaged transfer resistance of 21 s cm$^{-1}$ and a CH$_3$Cl background concentration of 540 pmol mol$^{-1}$ and the land use categories from Shorter et al. (1995), we obtain an upper limit of 1300 Gg a$^{-1}$ for $m_d$.

As depicted in figure 4, the apparent isotope effect of the soil uptake is bracketed by the isotope effects of both steps and decreases when increasing the net soil uptake. The microbial degradation is rate limiting at low net uptake rates, and the apparent isotope effect of the soil uptake is close to that for microbial degradation. For instance, a soil sink of 250 Gg a$^{-1}$ reveals an apparent ε value of -38 ‰. When the entire chloromethane diffusing into soils is microbially degraded, diffusion becomes the rate limiting step, and the apparent ε value matches that of diffusion.

In turn the imprint on the tropospheric $\delta^{13}$C shows a parabolic distribution with a maximum at $m_m = 0.5\ m_d$. The isolated effect of the soil sink would result in a maximum enrichment of 3.8 ‰ in the tropospheric $\delta^{13}$C that reduces to 2.1 ‰ when accounting for the concurrent reduction in the OH sink. In this case, increasing the soil sink could even lead to a decrease in the overall sink isotope effect once the apparent isotope effect of the soil sink becomes smaller than the isotope effect of the OH sink.

### 3.5 Seasonal variations in the $\delta^{13}C$ of tropospheric CH$_3$Cl

The OH driven seasonal cycle in the tropospheric mixing ratios of CH$_3$Cl implies an inverse co-variation in the $\delta^{13}C$ of tropospheric CH$_3$Cl to an extent that is closely linked to the kinetic isotope effect of the OH sink.

Our model resembles mean tropospheric mixing ratios of about 540 pmol mol$^{-1}$ (Monzka et al., 2010; Carpenter et al., 2014) and the seasonal cycles of CH$_3$Cl in both hemispheres within ±4 % (Fig. 2 upper panel). In our simulations, an $\epsilon$ value of -59‰ for the OH sink, produces an inverse co-variation of the $\delta^{13}C$(CH$_3$Cl) with the CH$_3$Cl mixing ratios with a seasonal amplitude of 9.2 ‰, whereas our new smaller $\epsilon$ value of -11.2 ‰ results in a seasonal amplitude of only 1.7 ‰ (Fig 2 lower panel) fitting quite well to the measured variation given by Thompson et al. (2002).

Random variations of ±10 ‰ in the isotopic source signatures, seasonal variations of the emission functions and variations in the soil sink resulted in seasonal fluctuations of up to ± 10 ‰ in the combined isotopic source signature. (see supplementary material for further details) As already noted by Tans (1997) the large tropospheric background strongly attenuates temporal variations. In our model simulations these seasonal variations in the combined isotopic source signature resulted in systematic seasonal variations of the northern hemispheric $\delta^{13}C$ of less than ± 0.7 ‰ attributable to the isotopic source signal and a scatter of up to ± 1.0 ‰ in the monthly mean $\delta^{13}C$(CH$_3$Cl) values in the northern hemisphere. In all our model simulations these variations in the combined isotopic source did not significantly affect the differences in the seasonal amplitude of the tropospheric $\delta^{13}C$ signal. They may have an imprint on the tropospheric $\delta^{13}C$(CH$_3$Cl) signal, when applying an $\epsilon$ value of -11.2 ‰ to the OH sink. However, they are largely obscured, when applying an $\epsilon$ value of -59 ‰ to the OH sink (Fig. S6 in the supplement). Masking the isotope effect of this large $\epsilon$ value would require seasonal variations by about 50 ‰ in the combined source signatures being inversely correlated to the OH sink. In sum, the lacking covariation between the mixing ratios and carbon isotope ratios strongly supports our new $\epsilon$ value of -11.2 ‰ and makes the previously reported larger $\epsilon$ value highly unlikely.

### 3.6 Implications for the tropical rainforest source

An $\epsilon$ value of -59 ‰ for the OH sink requires a mean mass weighted isotopic source composition of -84.5 ‰ to balance the tropospheric $\delta^{13}C$(CH$_3$Cl) of (-36.4 ± 2.1) ‰ (Thompson et al., 2002), as shown in previous studies. Apart from large emissions from higher plants (Keppler et al., 2005; Saito & Yokouchi, 2008) in tropical rainforests, this isotope effect suggests additional substantial emissions from an even more depleted source, such as senescent leaf litter (Keppler et al., 2005; Saito & Yokouchi, 2008). In contrast, the revised smaller $\epsilon$ value of -11.2 ‰ requires a mean isotopic source signature of -48.5 ‰, being close to the mass weighted $\delta^{13}C$ of all other known sources excluding higher plants. Along with higher plant emissions of 2200 Gg a$^{-1}$, the new $\epsilon$ value of -11.2 ‰ yields a mean tropospheric $\delta^{13}C$(CH$_3$Cl) of -56 ‰, being depleted by almost 20 ‰ in comparison to the mean reported tropospheric $\delta^{13}C$(CH$_3$Cl). We performed more than 10,000 steady state runs with random variations in the isotopic composition of tropospheric CH$_3$Cl (-36.4±2.1 ‰), the isotopic source signatures (table 4) and the isotope effect of the soil sink to assess the range of CH$_3$Cl-emissions from higher plants. The source category "Higher plants" was divided in two fractions, one representing "true" emissions from higher plants and the other representing missing emissions. The $\delta^{13}C$ of the missing emissions was always scaled to match the tropospheric $\delta^{13}C$(CH$_3$Cl).

As shown in Figure 5, the strength of the tropical rainforest source is directly linked to the strength and isotopic composition of missing emissions. A tropical rainforest source of (600 ± 200) Gg a$^{-1}$ suggests missing emissions of (1600 ± 200) Gg a$^{-1}$, requiring a $\delta^{13}C$ of (-45 ± 6) ‰ to balance the tropospheric $\delta^{13}C$(CH$_3$Cl). This $\delta^{13}C$ is close to

the mean isotopic composition of all other known sources. Increasing the emissions from all other known sources within the given ranges might reduce the missing emissions by about 500 Gg a$^{-1}$. A further increase of the tropical rainforest emissions results in an equivalent reduction in missing emissions but requires a more enriched $\delta^{13}$C for the missing emissions. For instance, balancing a tropical rainforest source of (1100 ± 200) Gg a$^{-1}$ requires missing

emissions of the same magnitude having a $\delta^{13}$C of (-31 ± 6) ‰. This is at the upper end of reported source signatures and may thus serve as a boundary to constrain the rainforest source from an isotopic perspective.

## 4    Carbon density based revision of the tropical rainforest source

Interestingly, support for our lower estimate arises from previous studies on the tropical rainforest CH$_3$Cl source when using above ground carbon density instead of coverage area for upscaling the CH$_3$Cl emission factors.

Large uncertainties in upscaling of local derived plant emissions to global scales can arise from i) temporal variations in the emissions, ii) spatial variability in environmental drivers, species composition and vegetation cover. Within the widely used FAO land cover classes, forests are defined as land with a tree cover exceeding 10 %, a potential tree height of 5 m and an area of at least 0.5 ha (FAO, 2012). Tropical rainforests encompass such sparsely covered areas with a carbon density of only a few Mg ha$^{-1}$ (Asner et al., 2010; Pereira Junior et al., 2016) as well as very dense

mature rainforests with a canopy height of more than 40 m and above ground carbon densities sometimes exceeding 300 Mg ha$^{-1}$ (Kato et al., 1978). This suggests a large variability in biomass that cannot be assessed with the previously used area based upscaling approaches. Area based estimates may be improved by leaf area index or above ground carbon density based approaches. There is some indication that CH$_3$Cl is mainly emitted by mature trees (Saito and Yokouchi, 2008; Saito et al., 2008; 2013). This is more readily reflected by carbon density than by leaf area. Further,

the available carbon density data products allow a direct discrimination between tropical forests and other tropical vegetation. We thus propose a carbon density based upscaling approach of experimentally derived emission factors to reduce uncertainties arising from the spatial variability in above ground biomass. We first convert reported area based emission factors to carbon density based emission factors and then multiply them with the carbon stock of the tropical rainforest:

$$F(CH_3Cl) \ = \ \frac{E_F * C_{RF}}{C_{st}} \hspace{10cm} (5)$$

Here, $F(CH_3Cl)$ is the source strength [Gg a$^{-1}$], $E_F$ is the experimentally derived emission factor [Gg ha$^{-1}$ a$^{-1}$], $C_{st}$ is the above ground carbon density assigned to the sampling site [Gg ha$^{-1}$] and $C_{RF}$ is the estimated total above ground biomass [Gg] of the respective biome, in this case the tropical rainforest.

The first direct evidence for strong CH$_3$Cl emissions from tropical plants came from branch incubations of tropical

plants in a greenhouse (Yokouchi et al., 2000). This study revealed particularly high emission from dipterocarp species being dominant in tropical lowland rainforests of South and Southeast Asia, and suggested mean CH$_3$Cl emissions of 74 µg m² h$^{-1}$. Several follow up studies carried out in tropical rainforests reported ten- to fivefold lower fluxes (Saito et al., 2008; 2013; Gebhardt et al., 2008; Blei et al., 2010). We exclude the high emission factor from the greenhouse study from our reanalysis of the tropical rainforest source and focus on the studies, providing experimentally derived

emission factors for CH$_3$Cl emissions from tropical forests and allowing for a sufficient estimate of carbon densities assignable to them.  Details on these studies are provided in table 5. Three studies have been carried out in lowland tropical rainforests of South East Asia, and one has been carried out over Surinam in South America. We are not aware

of any CH$_3$Cl flux measurements from African tropical rainforests. Two studies relied on branch or leaf incubation to measure CH$_3$Cl fluxes (Saito et al., 2013; Blei et al., 2010). A third study used a micrometeorological approach and in addition performed leaf and branch incubation (Saito et al., 2008). The remaining study (Gebhardt et al., 2008) derived CH$_3$Cl emissions factors from concentration gradients above the rainforest. The concentration gradients were obtained from canister samples taken at different heights above the rainforest from an airplane. The results from branch and leaf incubations were first normalized to leaf dry weight and then converted to area based emission factors using reported allometric data for Southeast Asian tropical lowland rainforests along with assumptions on the distribution and abundance of the investigated species. The mean area normalized fluxes (8.0 µg m² h$^{-1}$ ±45%) from these studies show a notably larger variability than the original leaf biomass normalized fluxes (0.028 µg g$^{-1}$ h$^{-1}$ ± 10%), although all three studies referred to the same allometric data (Yamakura et al., 1986) in their conversion. Noteworthy, the study reporting the lowest emissions factors from branch enclosures reported almost three times higher fluxes using a micrometeorological approach. In sum, the area based factors agree within a factor of 3 and range from 5.0 to 14 µg m² h$^{-1}$ (9.1 µg m² h$^{-1}$ ± 37%).

The three south East Asian studies refer to a dense and mature dipterocarp forest with an above ground carbon density of (265 ± 44) Mg C ha$^{-1}$ (Yamakura et al., 1986) that we apply here. For the study carried out above the rainforest of Surinam, we derived a carbon density of (160 ± 15) Mg ha$^{-1}$ from carbon density maps (Saatchi et al., 2011; Baccini et al., 2012) .This range agrees with the FAO estimate for French Guyana (FAO, 2015) and is supported by several field surveys carried out in this region (Chave et al., 2001; 2008). With this, we obtain a mean carbon density based emission factor of (4.0 ± 1.2) g Mg$^{-1}$, referring to a mean carbon density of 202 Mg ha$^{-1}$. This is well above the average tropical rainforest carbon density, ranging from 96 to 117 Mg ha$^{-1}$ (Bachini et al, 2012; Saatchi et al., 2011; Köhl et al., 2015; FAO, 2015).

In consequence, our carbon density based estimates of the tropical CH$_3$Cl source are 30 to 70 % lower than the respective area based estimates (table 6). The difference is in the range of 30 % for dense old grown evergreen forests such as the Tierra Firme forests of French Guyana, between 40 % and 50 % for the moist tropical rainforest, and increases to almost 70 % for the entire pantropical forests including dry tropical forests, degraded forests and plantations. This trend reflects the decreasing trend in carbon density in the tropical rainforest biomes as well as the effect of forest degradation. Regardless of the source for the carbon density estimates, our approach suggests a tropical rainforest CH$_3$Cl source of (670 ± 250) Gg a$^{-1}$, being 53 % to 65 % lower than the respective area based estimates in the range of 1200 to 2000 Gg a$^{-1}$ (Saito et al., 2008; 2013; Gebhardt et al., 2008; Blei et al., 2010).

The uncertainty in the area based emission factors is estimated to 24 % from the standard deviation of the reported means. Additional uncertainties for our carbon density based upscaling (as compared to the previous area based upscaling) arise from the uncertainties in the total above ground carbon stocks (±8.6 %) and the site specific carbon density (±15 %). Using error propagation, we estimate the total uncertainty of our approach to ±30.4 %. However, we note an urgent need for more detailed flux studies. Currently there is no information about how physiological and environmental drivers might affect CH$_3$Cl emissions from tropical rainforests. Apart from the observation that some members of the Dipterocarpacae family are particular strong emitters of CH$_3$Cl, this also holds true with respect to species composition.

## 5    Conclusions

We reported new ε values for the reaction of $CH_3Cl$ with OH and Cl of (-11.2 ± 0.8) ‰ (n=3) and (-10.2 ± 0.5) ‰ respectively being five to seven times smaller than the previous reported ε values for these reactions. Strong support for the reliability of our new fractionation factors arises from the absence of any significant co-variation in the mixing and carbon isotope ratios of tropospheric $CH_3Cl$.

Conjoining our new KIEs of the tropospheric $CH_3Cl$ sinks and the biomass based upscaling of previously reported emission factors suggest a tropical vegetation source of only (670 ± 200) Gg $a^{-1}$, being about threefold smaller than suggested in current budgets. We assign $\delta^{13}C$ of -45 ± 6 ‰ to the missing emissions of (1530 ± 200) Gg $a^{-1}$. Notably increasing the soil sink by 750 Gg $a^{-1}$ and decreasing biomass burning emissions by 460 Gg $a^{-1}$, as suggested in the latest assessment on ozone depleting substances (Carpenter et al., 2014), would substantially increase this gap but have a negligible effect on the isotopic composition of the missing emissions. The $\delta^{13}C$ value of the missing emissions matches with the mean source signature of the other known sources (except rainforests). Increasing these emissions within the given ranges might reduce the gap to (1100 ± 200) Gg $a^{-1}$. From a purely isotopic perspective, in particular larger emissions from biomass burning could further reduce this gap. However, this is highly speculative as virtually any source combination providing a mean $\delta^{13}C$ of -45 ± 6 ‰ could fill the gap.

With $CH_3Cl$ being the single largest natural carrier of chlorine to the stratosphere, predicting future baselines of stratospheric chlorine requires a better understanding of the global $CH_3Cl$ cycle and an identification of the missing emissions.

*Data availability.* The data used in this publication and the model code are available to the community and can be accessed by request to the corresponding author.

*Acknowledgements.* We acknowledge the German Federal Ministry of Education and Research (BMBF) for funding within SOPRAN 'Surface   Ocean Processes in the Anthropocene (grants 03F0611E and 03F0662E).  This study was further supported by DFG (KE 884/ 8- 1; KE 884/8- 2, KE 884/ 10- 1) and by the DFG  research unit  'Natural Halogenation Processes in the Environment - Atmosphere and    Soil' (KE 884/7- 1, SCHO 286/7- 2, ZE 792/5 -2). We finally thank, S. O'Doherty, and P.J. Fraser for the AGAGE data from Mace Head and Cape Grim. AGAGE is supported principally by NASA (USA) grants to MIT and SIO, and also by: DECC (UK) and NOAA (USA) grants to Bristol University; CSIRO and BoM (Australia): FOEN grants to EMPA (Switzerland); NILU (Norway); SNU (Korea); CMA (China); NIES (Japan); and Urbino University (Italy)

*Author contributions* E.B., F.K, J.W. and C.Z. designed the experiment, E.B, J.W and C.Z carried out the Smog chamber experiments. M.G carried out the isotope analysis in Heidelberg and E.B carried out the isotope analysis in Hamburg. E.B performed the modelling work.  All authors contributed equally to the preparation of the manuscript.

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

# 7 Tables

**Table 1: Experimental conditions of the degradation experiments with OH. The $O_3$ mixing ratios are average steady state mixing ratios throughout the experiment, the $Cl_2$ mixing ratios refer to the initial mixing ratios at the beginning of each photolysis sequence.**

| Exp. | reactant | | oxidant | $O_3$ $Cl_2$ | Irridiation | $H_2$ | rel. Hum. | T | OH |
|---|---|---|---|---|---|---|---|---|---|
| | | µmol mol$^{-1}$ | | µmol mol$^{-1}$ | | µmol mol$^{-1}$ | % | °C | cm$^{-3}$ |
| 1 &2 | CH$_3$Cl | 5 , 10 | OH | 0.62 | 1x55W, $\lambda_{max}$ = 254 nm | 2000 | 65 | 20.7 | 2.9 x 10$^9$ |
| 3 | CH$_3$Cl | 0.13 | OH | 0.62 | 1x55W, $\lambda_{max}$ = 254 nm | 2000 | < 1 | 20.6 | 8.7 x 10$^7$ |
| 4 | CH$_3$Cl, CH$_4$ | 13, 5 | OH | 3.7 | 4x55W, $\lambda_{max}$ = 254 nm | 2000 | 65 | 20.4 | 1.6 x 10$^{10}$ |
| 5 | CH$_4$ | 5 | | 0 | 4x55W, $\lambda_{max}$ = 254 nm | | 72 | 20.3 | |
| 6 | CH$_4$ | 6 | OH | 3.7 | 4x55W, $\lambda_{max}$ = 254 nm | | 72 to 75 | 20.3 | 1.6 x 10$^{10}$ |
| 7 | CH$_3$Cl | 10 | Cl | 2 to 10 | 7x1200W 300 - 700 nm | | < 1 | 20.7 | |
| 8 | CH$_4$ | 5 | Cl | 2 to 10 | 7x1200W 300 - 700 nm | | < 1 | 20.5 | |

**Table 2: Summary of the kinetic isotope effects (ε) for the reaction of CH3Cl and CH4 with OH and Cl from this study. We used an orthogonal regression to calculate ε and the respective uncertainties on the 1σ level for each experiment. In experiment 4 the ε for methane has not been determined.**

|  |  | Hamburg | | | Heidelberg | | |
|---|---|---|---|---|---|---|---|
|  |  | ε | | $R^2$ | | ε | $R^2$ |
| Exp. 1 | $CH_3Cl$ +OH | -12.1 | ± 0.6 | 0.95 | -11.7 | ± 0.4 | 0.99 |
| Exp. 2 |  | -12,1 | ± 0.3 | 0.99 | -10.5 | ± 0.3 | 0.99 |
| Exp. 4 |  | -10.4 | ± 0.4 | 0.99 | -10.6 | ± 0,6 | 0.99 |
| Exp. 7 | $CH_3Cl$ + Cl | -10.3 | ± 0.7 | 0.96 | -10.4 | ± 0.4 | 0.98 |
| Exp. 5 | $CH_4$ + OH |  |  |  | -4.7 | ± 0.2 | 0.99 |
| Exp. 8 | $CH_4$ + Cl |  |  |  | -59.0 | ± 1.3 | 0.99 |

**Table 3: Compilation of kinetic isotope effects (ε) for the reaction of CH3Cl, CH4 and alkanes with OH and Cl**

| Reaction | | | ε | | | Method | Reference |
|---|---|---|---|---|---|---|---|
| $CH_3Cl$ | + | OH | -58 | ± | 10 | smog chamber; $O_3+H_2+hv$ (254 nm), FTIR | Gola et al., 2005 |
|  |  |  | -44 |  |  | theoretical at 298K | Feilberg et al., 2005 |
|  |  |  | -3.6 |  |  | theoretical | Jalili, S. & Akhavan, 2006 |
|  |  |  | -5 | ± | 3 | derived from field data | Thompson et al., 2002 |
|  |  |  | **-11.2** | **±** | **0.8** | **smog chamber; $O_3+H_2O+hv$ (254 nm), GC-IRMS** | **This study** |
| $CH_3Cl$ | + | Cl | -70 | ± | 10 | smog chamber; $Cl_2+hv$ (370 nm), FTIR | Gola et al., 2005 |
|  |  |  | -35 |  |  | theoretical at 298K | Feilberg et al., 2005 |
|  |  |  | **-10.4** | **±** | **0.5** | **smog chamber; $Cl_2+hv$ GC-IRMS** | **This study** |
| $CH_4$ | + | OH | -3.9 | ± | 0.4 | photo reactor; $H_2O_2+hv$, GC-IRMS | Saueressig et al. 2001 |
|  |  |  | -5.4 | ± | 0.9 | photo reactor, $O_3+H_2O+hv$ (254 nm), GC-IRMS | Cantrell et al., 1990 |
|  |  |  | **-4.7** | **±** | 0.2 | **smog chamber; $O_3+H_2O+hv$ (254 nm), GC-IRMS** | **This study** |
| $CH_4$ | + | Cl | -58 | ± | 2 | smog chamber; $Cl_2+hv$ FTIR | Sellevåg et al., 2006 |
|  |  |  | -66 | ± | 2 | photo reactor; $Cl_2+hv$; TDLAS | Saueressig et al., 1995 |
|  |  |  | -62 | ± | 0.1 | smog chamber; $Cl_2+hv$; DI-IRMS | Tyler et al., 2000 |
|  |  |  | -59 | ± | 1.3 | **smog chamber; $Cl_2+hv$ GC-IRMS** | **This study** |
| $C_2H_6$ | + | OH | -7.5 | ± | 0.5 | reaction chamber; $H_2O_2$ + hv; GC-IRMS | Piansawan et al., 2017 |
| R-**CH3** | + | OH | -18.7 | ± | 5.2 | reaction chamber; $R-NO_2$, NO + hv; GC-IRMS | Anderson et al., 2004 |
| R-**CH3** | + | Cl | -18.6 | ± | 0.3 | reaction chamber; $Cl_2$ + hv; GC-IRMS | Anderson et al. 2007 |

**Table 4: Simplified CH3Cl source and sink scheme used in the model**

| sources | strength [Gg a$^{-1}$] | | | δ$^{13}$C (source) / ε (sink) [‰] | | |
|---|---|---|---|---|---|---|
| | best | range | | best | Range (1σ) | |
| Biomass burning | 910 | 655 - | 1125 | -47 | -40 - | -52 |
| Oceans | 335 | 210 - | 480 | -36 | -30 - | -42 |
| Higher Plants / unknown | 2400 | 0 - | 3095 | -83 | -70 - | -96 |
| Other known sources | 365 | 79 - | 1016 | -45 | -39 - | -51 |
| **sinks** | | | | | | |
| OH, Cl | 3614 | 3564 - | 3000 | -11.2 / -59 | | |
| soils* | 250 | 200 - | 1000 | -37 | -46 - | -2 |
| stratosphere | 146 | | | 0 | | |

* The apparent isotope effect of the soil sink depends on its strength. See text for more details

**Table 5: Calculation of carbon density based CH3Cl emission factors from previously reported area based emission factors.**

| Site | Method | Carbon density | Emission per | | | | Ref. |
|---|---|---|---|---|---|---|---|
| | | | leaf dry mass | area | | Carbon density | |
| | | Mg ha$^{-1}$ | µg g$^{-1}$h$^{-1}$ | µg m$^{-2}$h$^{-1}$ | g ha$^{-2}$ a$^{-1}$ | g Mg$^{-1}$ a$^{-1}$ | |
| Glass house, Japan | branch enclosure | 325 | 0.32 | 74,0 | | | Yokouchi et al., 2000 |
| Pasoh Forest Reserve, Malaysia | micrometerological branch enclosure | 265 | 0.03 | 14,0 5,0 | 1226 | 4.6 | Saito et al., 2013 |
| Pasoh Forest Reserve, Malaysia | branch enclosure | 265 | 0.026 | 7,0 | 615 | 2.3 | Saito et al., 2013 |
| Danum Valley, Borneo Borneo | branch enclosure | 265 | 0.03 | 12,0 | 1051 | 4 | Blei et al., 2010 |
| Surinam, French Guyana | gradient above canopy | 160 | | 9,5 | 832 | 5.2 | Gebhardt et al., 2008 |
| **mean** | | | | | **931** | **4** | |
| **SD** | | | | | **265** | **1.2** | |

**Table 6: Comparison of calculated area based and carbon density based emissions from tropical forests**

| Region | Area | C density | Area based | | | Carbon density based | | | |
|---|---|---|---|---|---|---|---|---|---|
| | $10^6$ ha | $10^6$ g ha$^{-1}$ | Gg a$^{-1}$ | | | Gg a$^{-1}$ | | | % of area based |
| Brazil | 586 | 112 [1] | 496 | ± | 143 | 247 | ± | 73 | 50 |
| Indonesia | 165 | 112 [1] | 140 | ± | 40 | 69 | ± | 20 | 50 |
| Congo | 205 | 92 [1] | 173 | ± | 50 | 71 | ± | 21 | 41 |
| Trop. Africa | 775 | 62 [1] | 655 | | 189 | 182 | ± | 53 | 28 |
| Trop America | 1209 | 77 [1] | 1022 | | 295 | 356 | ± | 103 | 35 |
| Trop Asia | 474 | 98 [1] | 401 | | 116 | 176 | ± | 51 | 44 |
| **Pantropics** | **2458** | **78 [1]** | **2079** | **±** | **601** | **720** | **±** | **212** | **35** |
| | | | | | | | | | |
| Trop. Africa | 393 | 82 [2] | 332 | ± | 96 | 121 | ± | 36 | 36 |
| Trop America | 788 | 116 [2] | 666 | ± | 192 | 343 | ± | 101 | 52 |
| Trop Asia | 289 | 119 [2] | 244 | ± | 71 | 129 | ± | 38 | 53 |
| **Pantropics** | **1470** | **105 [2]** | **1243** | **±** | **359** | **580** | **±** | **171** | **47** |

[1] Saatchi et al. , 2011; [2] Bacchini et al., 2012

## 8    Figures

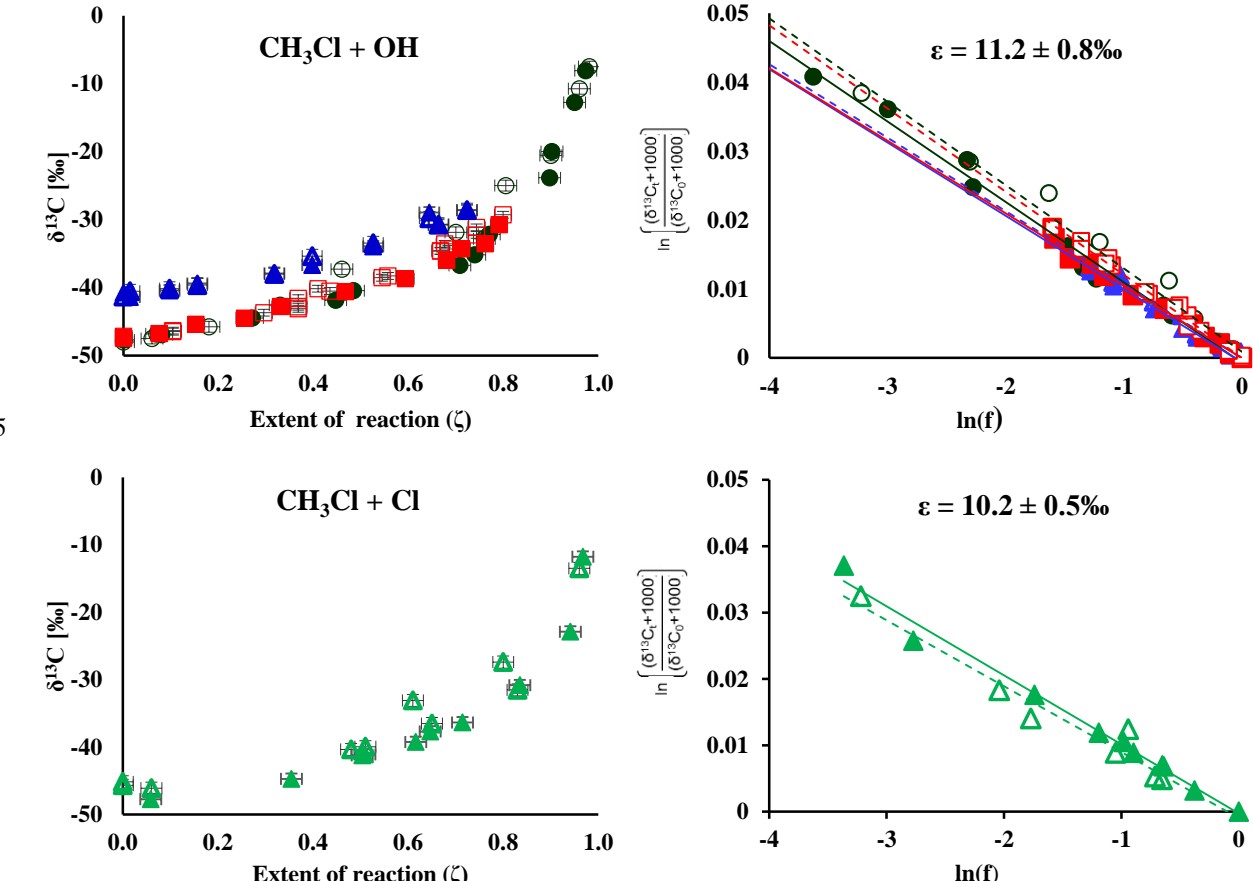

**Figure 1: Change in $\delta^{13}C$ over extent of reaction (left hand side) and corresponding Rayleigh plots (right hand) from the CH₃Cl degradation experiments. Filled symbols and regression lines refer to the data from Heidelberg, open symbols and dashed regression lines show data from Hamburg. The colors refer to different degradation experiments (black: experiment 1; red: experiment 2, blue: experiment 4 and green: experiment 7). Errors in ζ were ±2% on the 1σ level. Errors in $\delta^{13}C$, as derived from the regression analysis, ranged from ±0.4 to ±1.4‰ on the 1σ level.**

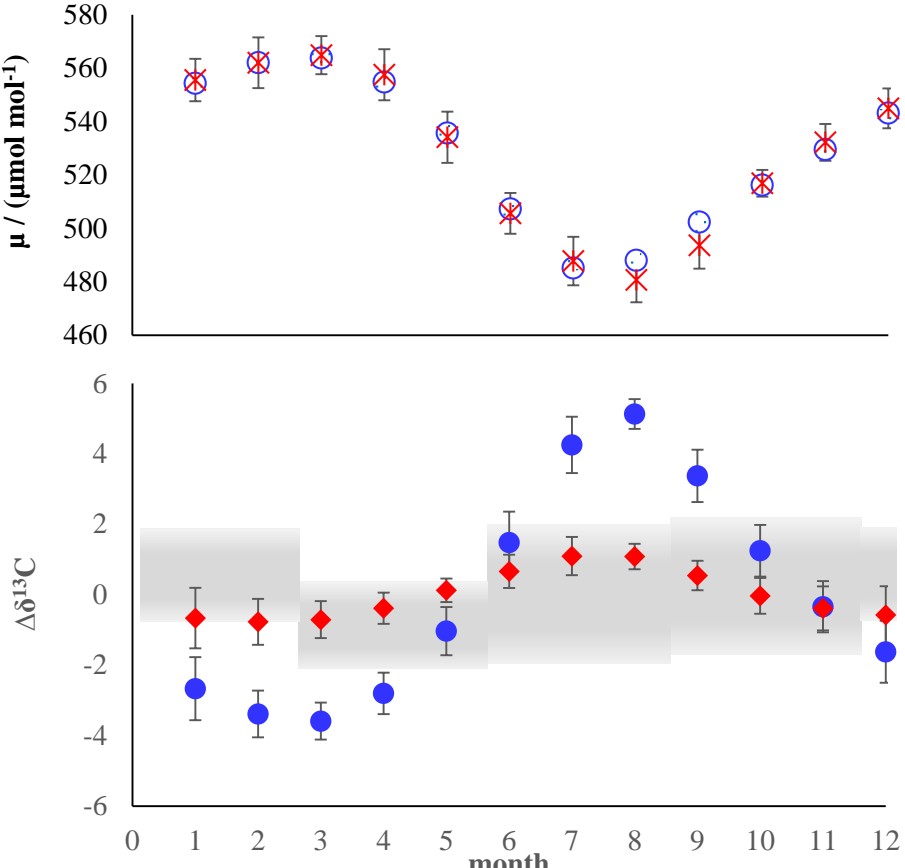

**Figure 2: Upper panel: Comparison of modeled Northern hemispheric mixing ratios (blue open diamonds) with measured mixing ratios at Mace Head, Ireland (red crosses) for the period from 2004 to 2014 (Prinn et al., 2000). Error bars indicate the variations in the monthly means on the 1 σ level. Lower panel: Modeled seasonal fluctuations in the δ$^{13}$C of northern hemispheric CH$_3$Cl using an ε of -11.2‰ (red filled diamonds) and an ε of -59‰ (blue filled dots) as reported by Gola et al. (2005). The panel shows monthly averages from a 10-year simulation with seasonal variations of up to ± 10 ‰ in the combined isotopic source signature. Error bars indicate the variations in the monthly means on the 1σ level. The grey shaded area shows reported seasonal variations (seasonal mean ± 1σ) from Thompson et al. (2002).**

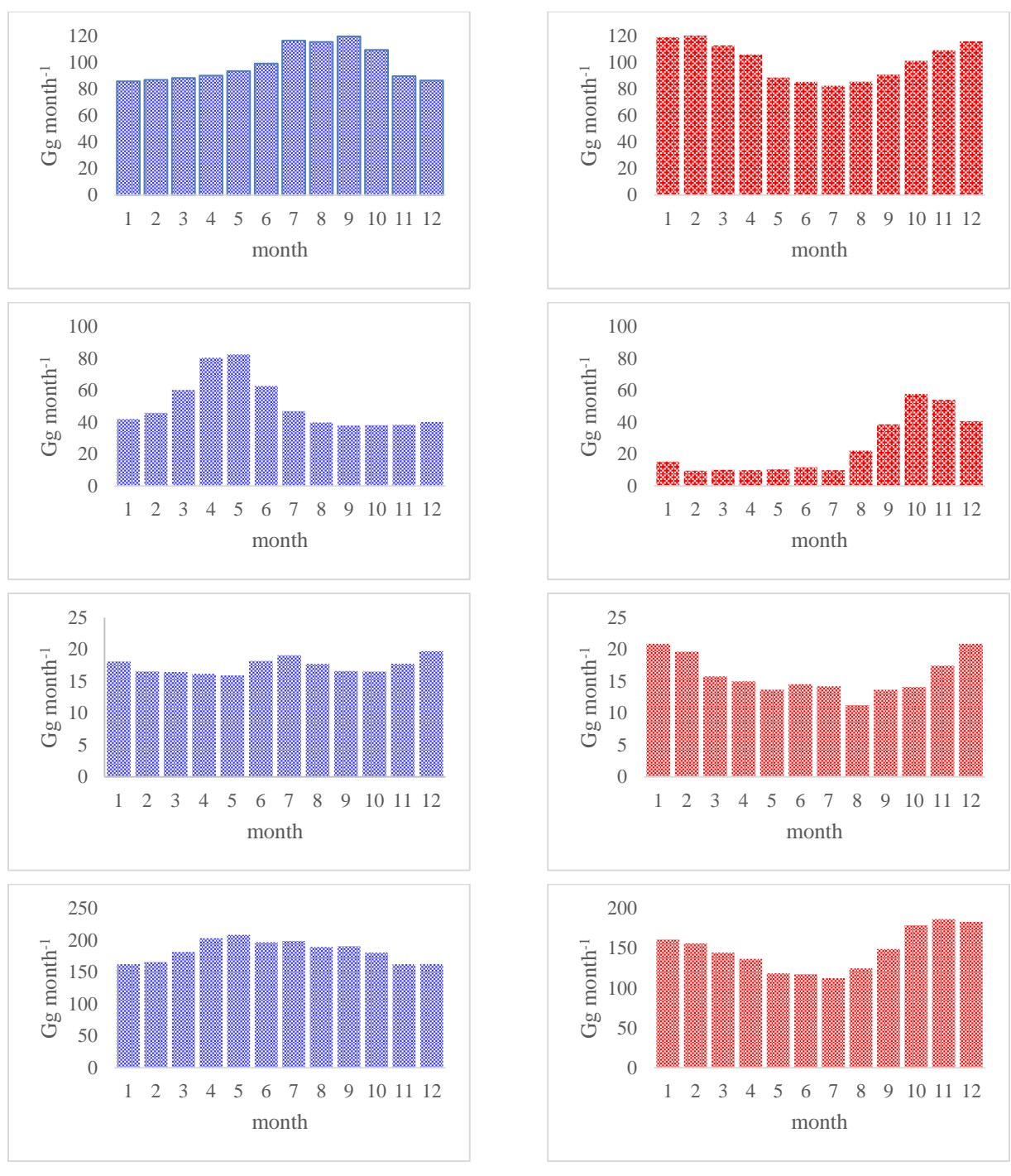

**Figure 3: Chloromethane emissions in Gg month⁻¹ for the northern (left side) and southern hemisphere (left side) Upper row panel: Combined emissions from higher plants and the unknown source; second row panel: biomass burning, third row panel: ocean net emission fluxes, fourth row panel total emissions. The emissions from the other known sources are constant over time with 22.8 Gg month⁻¹ in the northern hemisphere and 7.6 Gg month⁻¹ in the southern hemisphere.**

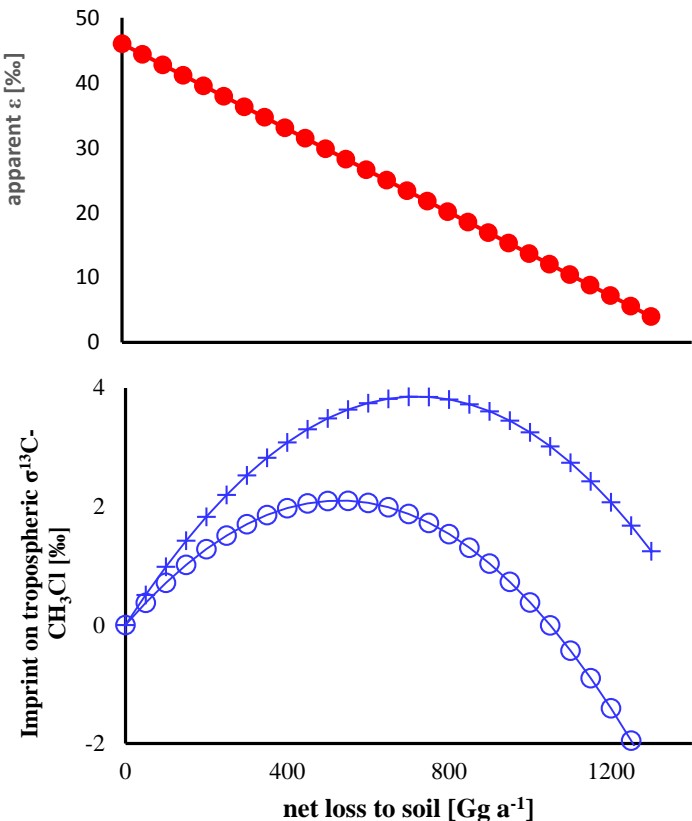

**Figure 4: The upper panel shows the apparent ε of the soil sink versus the sink strength. The lower panel shows the resulting effect on tropospheric δ¹³C. Crosses show the pure effect e.g. in the absence of any other fractionating sink. Open circles show the resulting effect with an ε of -11.2‰ assigned to the OH sink. Total losses were assumed to be fix at 4010 Gg a⁻¹.**

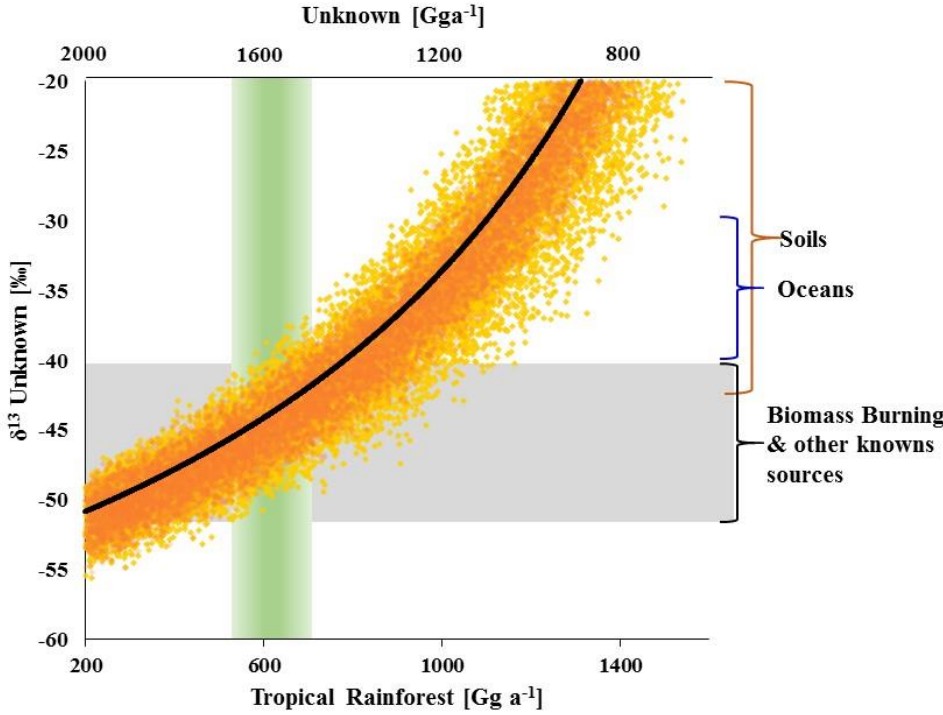

**Figure 5: Modeled isotopic composition of the missing source versus tropical rainforest emissions on the lower x-axis and missing emissions on the upper x-axis (rainforest = 2200 – unknown). The black line shows the best estimate derived from the mean isotopic source signatures. Orange dots indicate the range uncertainty (1σ) from uncertainties (1σ) in the $\delta^{13}$C of biomass burning (±7‰), ocean net emissions (±6‰) and other known sources (±6‰). Yellow dots mark the additional uncertainty from the $\delta^{13}$C of the tropical rainforest source (±10‰). The green column indicates the carbon density based estimate of the rainforest source and the grey bar indicates the range in $\delta^{13}$C of biomass burning and the mean from all sources excluding the tropical rainforest.**