# Peer review of "Evidence for a major missing source in the global chloromethane budget from stable carbon isotopes"

_Atmospheric Chemistry and Physics, 2018_

## Short Comment (SC1) · 17 Sep 2018

With regard to the missing source the authors might consider the following publication:

"Emission estimates of methyl chloride from industrial sources in China based on high frequency atmospheric observations", Li et al., Journal of Atmospheric Chemistry, 2017.

This might, in combination with the pollution transport mechanism explained in Oram et al., ACP, 2017, go some way to close the global chloromethane budget.

Best regards,

[Figure]

Johannes Laube

---

## Referee Comment (RC1) · M. S. Johnson (Referee) · 8 Oct 2018

General Comments. This paper presents convincing experimental evidence and arguments supporting the view that earlier measurements of the KIE of the $CH_3Cl + OH/Cl$ were in error, the true value being 5 to 6 times smaller. This is based on experiments carried out in a 3500 L smog chamber coupled with offline IRMS analysis of cannister samples. The analysis was further verified using GC-IRMS in two different laboratories. The conclusion is supported by the lack of a significant seasonal covariation in tropospheric $d13C(CH_3Cl)$ values. Further, comparisons are made with KIE's determined by experiment and theory in chemically similar systems, which would also seem

to indicate a revision in the accepted KIE. The revised KIE triggers an interesting and timely reanalysis of the atmospheric chloromethane budget.

Specific Comments. Accepting that other small molecules react with a smaller KIE, as argued at the bottom of page 6/top of page 7, this still does not account for the large KIE determined in the transition state theory study: As written in Sellevåg 2006, there will be a higher barrier to internal rotation in CH3Cl + OH than in CH4 + OH. The authors note that they cannot find a flaw in these studies by Sellevåg and Gola; Sellevåg may have agreed with the experimental work of Gola 2005 by a kind of perverse luck. In which case we should thank the authors of this work for their careful re-analysis. I would feel better though if there was a corresponding theoretical reanalysis; the question remains unanswered. CH3Cl + OH certainly will have a more constrained transition state than CH4 + OH.

Technical Corrections. I must complain a little bit about the use of units. 'micromole / mole' is fine, and I am often reading about 'ppm', but why use them both in the same paper? If you are going all in with IUPAC and SI notation then apply it consistently and use only the former and it's related forms, never the latter. (e.g. page 4 line 32, page 6 lines 11 and 12, etc.) Always put a space between number and unit (e.g. (page 1 line 22 = p1ln22), p4ln25, p4ln345, p11ln4, etc.) Are you using 'L' for the liter p1ln22 or 'l' p5ln33? Be consisitent. p3ln2, 'CMC Instruments' Check p3ln23, p5ln6, p6ln9

---

## Referee Comment (RC2) · J. Rudolph (Referee) · 11 Oct 2018

The paper presents measurements of the carbon kinetic isotope effects (KIE) for reactions of chloromethane OH-radicals and Cl-atoms. These reactions are the dominant loss processes for chloromethane in the atmosphere and knowledge of their KIE is essential for understanding the carbon isotope budget of atmospheric chloromethane. Measurements of the carbon KIE for reactions of chloromethane have been published before, but the measurements presented here hugely differ from those previously reported. This has a substantial impact on the use of chloromethane carbon isotope ratio measurements to constrain the sources of atmospheric chloromethane. The authors

also present a budget estimate for chloromethane based on their new KIE using a two box model. They conclude that there should be a major unknown source for atmospheric chloromethane in order to explain the atmospheric concentration and isotope ratio of chloromethane. Since chloromethane is the dominant natural source for stratospheric Cl it plays a major role in the budget of stratospheric ozone. Therefore, the subject of the paper is highly relevant for ACP. Overall the experimental data are well presented and sound. The two box model clearly demonstrates that, using the revised KIE, previously published chloromethane budgets no longer are consistent with the known atmospheric isotope ratios of chloromethane. Consequently. the paper should be published with some minor revisions. The authors argue that the low seasonal variability of the chloromethane carbon isotope ratio supports the finding of a much lower KIE for the atmospheric loss reactions of chloromethane than previously reported. To some extent I agree and it is indeed difficult to reconcile a huge isotope effect for chloromethane loss reactions with a strong seasonal variability of the concentration, but a small seasonal change of the atmospheric chloromethane carbon isotope ratio. The authors argue that based on the revised KIE a substantial unidentified source for chloromethane must exists. Since by definition the seasonal variability of the carbon isotope ratio for an unknown source is not known, there is some risk of circular reasoning resulting from assumptions of the strength of seasonal variability of emissions. For most sources there is very little direct information about the seasonality of the carbon isotope ratios of chloromethane emissions and for quite e few sources even magnitude and seasonality of emissions have large uncertainties. The plausibility that the seasonal variability of the strength and isotope ratios of chloromethane emissions nearly exactly balances the otherwise expected high seasonal variability of the carbon isotope ratio of chloromethane is a matter of debate, but cannot be completely dismissed. It also has to be remembered that recent estimates of the atmospheric chloromethane, including budgets which serve as basis for the two box models, budget have been influenced by the necessity to include a source for highly depleted chloromethane in order to reconcile atmospheric observations and the huge carbon KIE for atmospheric

removal reactions reported previously. In summary, the authors make a very strong point that the currently existing atmospheric chloromethane budgets are not consistent with the new carbon KIE for reactions of chloromethane. However, they should add some caveats that considering uncertainties in a budget without constraints from isotope ratio measurements will have extremely large uncertainties. In my opinion the strongest argument that the reported low KIEs are correct is the presented experimental evidence demonstrating the high quality of the measurements. The indirect arguments about better agreement with atmospheric observations weakens the main point. Based on the strong evidence for measurements of high quality it is extremely probable that there is s serious gap in the current understanding of the atmospheric budget of chloromethane. There are some minor details that need to be addressed: 1. KIE Measurements. Overall the description and presentation of the experiments and results are sound and demonstrate that the measurements are state of the art. I am aware that more details are presented in the cited paper by Keppler et al. (2018). However, given the fact that previously published measurements are very different from the results presented here, the authors should provide as much detail about the experiments and their results as possible. This could be done in a supplement to avoid adding length and material to the paper which would only be relevant for experts in laboratory studies of isotope effects. 1.1: Some more information about the linear range of the carbon isotope ratio measurements. Does the linear range cover the range of concentrations in the experiments should be provided? 1.2: Is there some explanation why the GC-IRMS measurements of the carbon isotope ratio of chloromethane in the artificial test mixture differs by 1.1 ‰ from the DI measurements? Could this be a linearity problem? The difference seems to be larger that the uncertainty of the measurements and larger than discrepancies reported previously in similar studies, including chloromethane. 1.3: The use of orthogonal regression to determine the KIE implies that the error of both variables have identical variances. How realistic is this. Does a conventional linear regression result in identical KIEs? 1.4 Based on the relative rates for the CH3Cl and CH4 reactions it should be possible to calculate a limit for

contributions from other possible reactions contributing to loss reactions of CH3Cl.

2. Uncertainty estimates (Page 10, lines 21-31, Figure 5): 2.1. Monte Carlo type calculations are very useful to estimate uncertainty ranges, but the results are not always easy to interpret in terms of the probability for a given value being within or outside of a given probability range. Does the scatter shown in Figure 5 represent a $\pm 1\ \sigma$ uncertainty range, a given percentile, or even a firm boundary (which would be difficult to interpret)? Some more explanations are needed. 2.2. Figure 5 presents uncertainties stemming from uncertainty in carbon isotope ratios of emissions. However, the magnitude of emissions form known sources also has substantial errors. Figure 5 only considers errors of isotope ratios of emissions. I understand that a "missing source" in principle is to some extent equivalent to underestimating identified sources. Still, some more explanation is needed to which extent uncertainties in the magnitude of identified sources may reduce the gap between identified sources and the unknown source.

3. Since current thinking seems to be that the tropical chloromethane emissions are from higher plants, the very simple extrapolations using biomass or rainforest areas may have extremely large uncertainties. For isoprene and terpenes many very detailed studies with the purpose of developing emission algorithms have been conducted and it has been clearly demonstrated that using biomass or forest area alone is clearly insufficient.

Technical details:

Page, line 1,21: remove e.g. 1,23 (and other locations): 3500 dm3 1, 28 (and other locations): Avoid inconsistencies in use of isotope fractionation, fractionation factor, kinetic isotope effect (definitions page 4, lines10-14). 1,30: remove tropical, I am not sure how well a two box model can support a specific unknown "tropical" source. 3, 27: 2 dm3 3, 27: baked out at which temperature? 3, 28: packing and dimensions of adsorption tubes should be provided. 3,35-37: Significant digits are inconsistent. Based on the 3 significant digits given for the working gas and reference standard, 4

significant digits for the results cannot be justified. 4,2: Notwithstanding? 4, 5: Based on error propagation the 1 $\sigma$ error of measurements the difference should be $\pm 0.8$ ‰ ate the 1 $\sigma$ level, which is somewhat lower than the observed value. This is not surprising, but still some comment should be added. 4, lines 10-15, eq. 1 and 2: no need to introduce $\alpha$, it is not used anywhere else in the paper. 4, 24: Table 1 4, 30: Prior to 4, 13 and 5, 6: Provide value for limit of "measureable loss" 5,22: . . .measured KIE. 5,24: replace "reasonably well" by "within. . ..", the possibility of bias due to impact from other reactions should be briefly discussed (see above, 1.4) either here or in 2.3. A higher OH-KIE may be due to interference from Cl-atom reactions. 5,33:.. whereas we used.. 6,1: "agree well" is subjective, the strongest argument against problems with mixing would be a presentation of the measured dilution factor combined with a comparison with a theoretical (volume and volume flow based), maybe in a supplement. 6,24: prior to 7, 33-34: Keppler et al. (2005) give (Table 1) a value of -38 ‰ with an uncertainty range of 4 ‰.8, 22-23: Needs more discussion. In a unidirectional emission the isotope ratio of the emissions is independent of atmospheric concentration and isotope ratio. In a bidirectional exchange atmospheric concentration and isotope ratio will influence the isotope ratio as well as the source strength of the net emissions. 8,25-30: "The lifetimes where then forced to. . ." is unclear and, if taken literally, questionable. In order to "force the lifetime" to reproduce the seasonal variations at Mace Head assumptions about the seasonality of emission rates are needed. Apart from the very limited a priori information on known sources, this would create circular reasoning about the seasonality of the "unknown source". 9,11: spore space 9,31: "nicely resembles"? 10,11: This conclusion depends strongly on assumption about the seasonal variation of the isotopic signature of emissions.

References should be checked carefully, I noticed that some cited publications are not included in the references. Table 4: The meaning of range should be explained (a $\pm$ n*$\sigma$ probability range, a percentile, a firm limit based on a given probability?)
* * *
* * *
Interactive
comment

---

## Author Comment (AC1) · 29 Oct 2018

The missing emissions and the tropical rainforest source sum up to 2200 Gg a-1 as shown in figure 5. With a revised tropical rainforest source of 670 Gg a-1, we obtain a missing source of 1530 Gg a-1. This correct value is provided on page 12, line 31. On page 1, line 30 and page 10, line 28 we provided values of 1230 Gg a-1 and 1400 Gg a-1. Moreover the stated uncertainties were not consistent. We apologize for this and will correct it during revision.

On behalf of all co-authors Enno Bahlmann

---

## Author Comment (AC2) · 5 Jan 2019

**Point by point reply to M. Johnson** (Referee comment are in italics)**:** We would like to thank M. Johnson for the positive evaluation of our manuscript and for the helpful comments to improve the manuscript. Requested changes were taken into account.

*General Comments. This paper presents convincing experimental evidence and arguments supporting the view that earlier measurements of the KIE of the $CH_3Cl + OH/Cl$ were in error, the true value being 5 to 6 times smaller. This is based on experiments carried out in a 3500 L smog chamber coupled with offline IRMS analysis of canister samples. The analysis was further verified using GC-IRMS in two different laboratories. The conclusion is supported by the lack of a significant seasonal covariation in tropospheric $d^{13}C(CH_3Cl)$ values. Further, comparisons are made with KIE's determined by experiment and theory in chemically similar systems, which would also seem to indicate a revision in the accepted KIE. The revised KIE triggers an interesting and timely reanalysis of the atmospheric chloromethane budget.*
*Specific Comments. Accepting that other small molecules react with a smaller KIE, as argued at the bottom of page 6/top of page 7, this still does not account for the large KIE determined in the transition state theory study: As written in Sellevåg 2006, there will be a higher barrier to internal rotation in $CH_3Cl + OH$ than in $CH_4 + OH$. The authors note that they cannot find a flaw in these studies by Sellevåg and Gola; Sellevåg may have agreed with the experimental work of Gola 2005 by a kind of perverse luck. In which case we should thank the authors of this work for their careful re-analysis. I would feel better though if there was a corresponding theoretical reanalysis; the question remains unanswered. $CH_3Cl + OH$ certainly will have a more constrained transition state than $CH_4 + OH$.*

**Authors reply:** We fully agree with M. Johnson that a theoretical reanalysis of the isotope effects for the reaction of small molecules (c1 to c3) with OH and Cl is highly desirable for improving our understanding of these isotope effects. Our approach was clearly an experimental one and unfortunately such a reanalysis cannot be done within our group.

*Technical Corrections. I must complain a little bit about the use of units. 'micromole / mole' is fine, and I am often reading about 'ppm', but why use them both in the same paper? If you are going all in with IUPAC and SI notation then apply it consistently and use only the former and it's related forms, never the latter. (e.g. page 4 line 32, page 6 lines 11 and 12, etc.) Always put a space between number and unit (e.g. (page 1 line 22 = p1ln22), p4ln25, p4ln345, p11ln4, etc.) Are you using 'L' for the liter p1ln22 or 'l' p5ln33? Be consistent. p3ln2, 'CMC Instruments' Check p3ln23, p5ln6, p6ln9.*

**Authors reply:** We will use the IUPAC and SI notation throughout the manuscript, thoroughly check all units in the manuscript and use $dm^3$ for litre.

---

## Author Comment (AC3) · 5 Jan 2019

**Point by point reply to J. Rudolph** (Referee comment are in italics): We would like to thank the referee for the positive and very constructive evaluation of our manuscript and for the helpful comments to improve the manuscript. Requested changes were taken into account.

*The paper presents measurements of the carbon kinetic isotope effects (KIE) for reactions of chloromethane OH-radicals and Cl-atoms. These reactions are the dominant loss processes for chloromethane in the atmosphere and knowledge of their KIE is essential for understanding the carbon isotope budget of atmospheric chloromethane. Measurements of the carbon KIE for reactions of chloromethane have been published before, but the measurements presented here hugely differ from those previously re- ported. This has a substantial impact on the use of chloromethane carbon isotope ratio measurements to constrain the sources of atmospheric chloromethane. The authors also present a budget estimate for chloromethane based on their new KIE using a two box model. They conclude that there should be a major unknown source for atmospheric chloromethane in order to explain the atmospheric concentration and isotope ratio of chloromethane. Since chloromethane is the dominant natural source for stratospheric Cl it plays a major role in the budget of stratospheric ozone. Therefore, the subject of the paper is highly relevant for ACP. Overall the experimental data are well presented and sound. The two box model clearly demonstrates that, using the revised KIE, previously published chloromethane budgets no longer are consistent with the known atmospheric isotope ratios of chloromethane. Consequently the paper should be published with some minor revisions. The authors argue that the low seasonal variability of the chloromethane carbon isotope ratio supports the finding of a much lower KIE for the atmospheric loss reactions of chloromethane than previously reported. To some extent I agree and it is indeed difficult to reconcile a huge isotope effect for chloromethane loss reactions with a strong seasonal variability of the concentration, but a small seasonal change of the atmospheric chloromethane carbon isotope ratio. The authors argue that based on the revised KIE a substantial unidentified source for chloromethane must exists. Since by definition the seasonal variability of the carbon isotope ratio for an unknown source is not known, there is some risk of circular reasoning resulting from assumptions of the strength of seasonal variability of emissions. For most sources there is very little direct information about the seasonality of the carbon isotope ratios of chloromethane emissions and for quite e few sources even magnitude and seasonality of emissions have large uncertainties. The plausibility that the seasonal variability of the strength and isotope ratios of chloromethane emissions nearly exactly balances the otherwise expected high seasonal variability of the carbon isotope ratio of chloromethane is a matter of debate, but cannot be completely dismissed. It also has to be remembered that recent estimates of the*

*atmospheric chloromethane, including budgets which serve as basis for the two box models, budget have been influenced by the necessity to include a source for highly depleted chloromethane in order to reconcile atmospheric observations and the huge carbon KIE for atmospheric removal reactions reported previously. In summary, the authors make a very strong point that the currently existing atmospheric chloromethane budgets are not consistent with the new carbon KIE for reactions of chloromethane. However, they should add some caveats that considering uncertainties in a budget without constraints from isotope ratio measurements will have extremely large uncertainties. In my opinion the strongest argument that the reported low KIEs are correct is the presented experimental evidence demonstrating the high quality of the measurements. The indirect arguments about better agreement with atmospheric observations weakens the main point. Based on the strong evidence for measurements of high quality it is extremely probable that there is a serious gap in the current understanding of the atmospheric budget of chloromethane.*

**Authors reply:** Our conclusion "that based on the revised KIE a substantial unidentified source for chloromethane must exist" is derived from yearly averaged emission and isotope data. Hence we don't think that this is a circular reasoning. We agree with J. Rudolph, that seasonal variations in the isotopic composition of the $CH_3Cl$ emissions are not known. For this reason, the isotopic source signatures of the CH3Cl emissions were randomly varied. On global and hemispheric scales seasonal variations in the isotopic source signatures have a surprisingly small effect on the tropospheric $\delta^{13}C(CH_3Cl)$, mainly because of the strong dilution effect of the tropospheric background. In our simulations seasonal variations in the combined isotopic source signature of about ±4 ‰ induced seasonal fluctuations in the tropospheric $\delta^{13}C(CH_3Cl)$ of less than 1.1 ‰. Seasonal variations in the range of ±7 to ±11 ‰ in the combined source signature, covering the reported uncertainty ranges of most source signatures result in seasonal variations of less than 3‰. Such variations still have a minor effect on the seasonality of the tropospheric $\delta^{13}C(CH_3Cl)$ when applying a KIE of -59 ‰ to the OH sink. Masking such a large isotope effect for the losses would require variations by about 50 ‰ in northern hemispheric emissions and seasonal variations of 27 ‰ in the southern hemispheric emissions. Moreover these large variations in the source signatures have to be inversely correlated with OH loss in both hemispheres.

To our opinion this is an extremely unlikely scenario and the good agreement between the modeled seasonal variations of the tropospheric $\delta^{13}C(CH_3Cl)$ with atmospheric observations hence strengthens our experimental results. As we could not detect any obvious flaws in the

studies of Gola et al. (2005) and Sellevåg et al (2006) we found it important to provide this additional evidence for our smaller KIEs.

*There are some minor details that need to be addressed:*

*1. KIE Measurements. Overall the description and presentation of the experiments and results are sound and demonstrate that the measurements are state of the art. I am aware that more details are presented in the cited paper by Keppler et al. (2018). However, given the fact that previously published measurements are very different from the results presented here, the authors should provide as much detail about the experiments and their results as possible. This could be done in a supplement to avoid adding length and material to the paper which would only be relevant for experts in lab- oratory studies of isotope effects.*

**Authors reply:** A more detailed description has been added to the supplementary information (Supplement S1).

*1.1: Some more information about the linear range of the carbon isotope ratio measurements. Does the linear range cover the range of concentrations in the experiments should be provided?*

**Authors reply:** The sampled $CH_3Cl$ amounts varied between 0.8 and 15 nmole. The Hamburg method has been shown to be linear from 0.01 to 20 nmole but with a decreasing reproducibility below 0.1 nmole (Bahlmann et al., 2011). The Heidelberg method was linear within the whole range of measured sample amounts. We will add this to the method section.

*1.2: Is there some explanation why the GC-IRMS measurements of the carbon isotope ratio of chloromethane in the artificial test mixture differs by 1.1 ‰ from the DI measurements? Could this be a linearity problem? The difference seems to be larger than the uncertainty of the measurements and larger than discrepancies reported previously in similar studies, including chloromethane.*

**Authors reply:** Indeed the difference between the DI and GC-C-IRMS is at the upper end of those reported in previous studies. The reason for this has not been exploited. As outlined above this has no effect on the linear range of the Hamburg method.

*1.3: The use of orthogonal regression to determine the KIE implies that the error of both variables have identical variances. How realistic is this. Does a conventional linear regression result in identical KIEs?*

**Authors reply:** For each experiment the KIEs obtained with an orthogonal regression agreed well with those obtained with a linear regression. For the reaction of $CH_3Cl$ with OH the orthogonal regression results in a KIE of (-11.2 ±0.8) ‰ and the linear regression results in a KIE of (-11.0 ± 0.7) ‰. For the reaction $CH_3Cl$ with Cl the respective KIEs were (-10.2 ± 0.5) ‰ and (-10.1 ± 0.5) ‰.

*1.4 Based on the relative rates for the CH3Cl and CH4 reactions it should be possible to calculate a limit for contributions from other possible reactions contributing to loss reactions of CH3Cl.*

**Authors reply:** The relative rate $CH_4/CH_3Cl$ in experiment 4 of 5.8 is 4.4 % larger than the recommended rate constants of 5.6 (Burkholder et al., 2015). The difference of 4.4 % might indicate small contributions from other loss processes but is within the stated uncertainties of the relative rate constants of 10% for $CH_4$ and $CH_3Cl$, respectively. The control experiment in the absence of water vapor revealed a loss of less than 3% over 10 h that can most likely be attributed to the reaction with OH originating from the reaction of $O^1D$ with $H_2$. In any case this experiment provides an upper limit of 3% for contributions from other possible reactions to the observed loss of $CH_3Cl$.

*2. Uncertainty estimates (Page 10, lines 21-31, Figure 5):*

*2.1. Monte Carlo type calculations are very useful to estimate uncertainty ranges, but the results are not always easy to interpret in terms of the probability for a given value being within or outside of a given probability range. Does the scatter shown in Figure 5 represent a ± 1 σ un- certainty range, a given percentile, or even a firm boundary (which would be difficult to interpret)? Some more explanations are needed.*

**Authors reply:** The scatter represents a ±1 σ uncertainty range with ±1σ uncertainties of 7 ‰ for biomass burning, 6 ‰ for ocean net emissions and other known sources and 10‰ for the tropical rainforest source. We have added this information to the revised manuscript in table 4 and figure 5.

*2.2. Figure 5 presents uncertainties stemming from uncertainty in carbon isotope ratios of emissions. However, the magnitude of emissions form known sources also has substantial errors. Figure 5 only considers errors of isotope ratios of emissions. I understand that a "missing source" in principle is to some extent equivalent to underestimating identified*

*sources. Still, some more explanation is needed to which extent uncertainties in the magnitude of identified sources may reduce the gap between identified sources and the unknown source.*

**Authors reply:** With respect to the modelling uncertainties in the carbon isotope ratio of emissions are to some extent equivalent to uncertainties in their source strengths.

At the current level of uncertainty in the isotopic source signatures all emissions except those from higher plants can be scaled to the upper limits of source strength within the stated uncertainties. We indicated this on page 10, line 29. On page 12 line 36 we further stated that increasing the emissions from these other known sources within the stated uncertainties could reduce the missing emissions from $(1530\pm190)$ Gg a$^{-1}$ to $(1100\pm200)$ Gg a$^{-1}$. We will clarify this issue in the revised manuscript. :

*3. Since current thinking seems to be that the tropical chloromethane emissions are from higher plants, the very simple extrapolations using biomass or rainforest areas may have extremely large uncertainties. For isoprene and terpenes many very detailed studies with the purpose of developing emission algorithms have been conducted and it has been clearly demonstrated that using biomass or forest area alone is clearly insufficient.*

**Authors reply:** We fully agree with the referee, that using biomass or forest area alone has its limitations regarding a global upscaling. Nevertheless, any upscaling can only be made on the basis of available information. Currently there is no information about how physiological and environmental drivers might affect $CH_3Cl$ emissions from tropical rainforests. Apart from the observation that some members of the Dipterocarpacae family are particular strong emitters of $CH_3Cl$, this also holds true with respect to species composition.

It should further be noted that the area bases on the previous bottom-up estimates are not consistent. Furthermore, Blei et al. (2008) and Saito et al. (2008; 2013) used an unrealistic high leaf biomass of 900 g m² in their flux conversion from leaf biomass to m². Given this, we found a simplified biomass based upscaling approach more reliable than an area based upscaling. However, we will mention the limitations of this approach and further suggest the need for more detailed $CH_3Cl$ flux studies in the tropics.

*Technical details:*
*Page, line 1,21: remove e.g.*

**Authors reply:** Change applied.

*1,23 (and other locations): 3500 dm3*

**Authors reply:** We now use dm$^3$ instead of litre throughout the whole manuscript.

*1, 28 (and other locations): Avoid inconsistencies in use of isotope fractionation, fractionation factor, kinetic isotope effect (definitions page 4, lines10-14).*

**Authors reply:** We checked the terminology for consistency.

*1,30: remove tropical, I am not sure how well a two box model can support a specific unknown "tropical" source.*

**Authors reply:** The support for an unknown tropical source mainly comes from previous 2- and 3-D model studies. However, "tropical" is an unnecessary restriction and will be removed.

*3, 27: 2 dm3*

**Authors reply:** We now use $dm^3$ instead of litre throughout the whole manuscript.

*3, 27: baked out at which temperature?*

**Authors reply:** The stainless steel canisters were heated at 250°C. The information has been added to the revised manuscript

*3, 28: packing and dimensions of adsorption tubes should be provided.*

**Authors reply:** The requested information has been added to the revised manuscript.

*3,35-37: Significant digits are inconsistent. Based on the 3 significant digits given for the working gas and reference standard, 4 significant digits for the results cannot be justified.*

**Authors reply:** Change applied.

*4,2: Notwithstanding?*

**Authors reply:** The sentence has been changed to read "A combustion reactor filled with copper (II) oxide at 850°C was used to convert $CH_3Cl$ and $CH_4$ into $CO_2$."

*4, 5: Based on error propagation the 1 σ error of measurements the difference should be ± 0.8 ‰ at the 1 σ level, which is somewhat lower than the observed value. This is not surprising, but still some comment should be added.*

**Authors reply:** This larger difference may result from small additional error of scales adding to the measurement uncertainties. For the purpose of this study we made no efforts to compare

the working standards used in both labs. We have added a short comment to the revised manuscript.

*4, lines 10-15, eq. 1 and 2: no need to introduce α , it is not used anywhere else in the paper.*
**Authors reply:** We removed "α" from eq. 1 and 2 in the revised manuscript.

*4, 24: Table 1 4, 30: Prior to 4, 13 and 5, 6: Provide value for limit of "measureable loss"*
**Authors reply:** The limit for a measurable loss was between 1.4 and 2% in our experiments. This information has been added to the manuscript.

*5,22: ... measured KIE.*
**Authors reply:** Change applied.

*5,24: replace "reasonably well" by "within ... .", the possibility of bias due to impact from other reactions should be briefly discussed (see above, 1.4) either here or in 2.3. A higher OH-KIE may be due to interference from Cl-atom reactions.*
**Authors reply:** "reasonably well" has been replaced by "within ... .". We added a brief discussion on potential interferences from side reactions to the revised manuscript. to the revised manuscript: "Prior to the methane degradation experiment (exp. 6 in table 1) we made the methane blank experiment (exp 5 in table 1) with no ozone but the UV-light being switched on. This experiment revealed no methane loss over 10 h. With this we can exclude any interferences from reactive chlorine during the $CH_4$-OH experiment. The higher KIE found here might result from the reaction of methane with $O^1D$. Cantrell et al. (1990), who also used UV-photolysis in the presence of water as an OH source, found an even higher KIE of (-5.4 ±0.9) ‰ and estimated that the reaction of $CH_4$ with $O(^1D)$ showing a KIE of -13 ‰ (Saueressig et al., 2001) may have contributed about 3% to the overall degradation."

*5,33:.. whereas we used..*
**Authors reply:** Change applied.

*6,1: "agree well" is subjective, the strongest argument against problems with mixing would be a presentation of the measured dilution factor combined with a comparison with a theoretical (volume and volume flow based), maybe in a supplement.*

**Authors reply:** The effect of mixing on the observed KIE can be approximated from the time scales of mixing and reaction according to the following equation (Morgan et al., 2004; Kaiser et al., 2006) :

$$\epsilon_{obs} \approx \frac{1}{2}\epsilon_i \times \left(1 + \sqrt{\frac{1}{1+Q}}\right)$$

Here $\epsilon_i$ is the intrinsic fractionation factor, $\epsilon_{obs}$ is the observed fractionation factor and Q is the ratio of the mixing time and reaction time scale (1/k). With a reaction time scale of at least 300 minutes and a mixing time scale of 10 minutes we obtain

$$\epsilon_{obs} \approx 0.99 \times \epsilon_i$$

This clearly shows that mixing problems are no issue in our study. We have added this information to the revised manuscript.

*6,24: prior to 7, 33-34: Keppler et al. (2005) give (Table 1) a value of -38 ‰ with an uncertainty range of 4 ‰ .*

**Authors reply:** We apologize for this shortened description. The value given in Keppler et al. (2005) served as a starting point in our study. It refers to Kommatsu (2004) reporting a mean $\delta^{13}C$ of -38‰ for CH3Cl in coastally influenced waters off Japan and more enriched $\delta^{13}C$ values in the range of -12‰ to -30‰ from the open North-East Pacific. We obtained average $\delta^{13}C$ values of  -43 ±3 ‰ from a productive lagoon in southern Portugal (Weinberg et al. 2014). Taking the biotic and abiotic degradation of CH3Cl into account we estimate the mean isotopic source signature of the ocean source to -36 ±6 ‰. We have added this information to the revised manuscript.

*8, 22-23: Needs more discussion. In a unidirectional emission the isotope ratio of the emissions is independent of atmospheric concentration and isotope ratio. In a bidirectional exchange atmospheric concentration and isotope ratio will influence the isotope ratio as well as the source strength of the net emissions.*

**Authors reply:** Saito et al. (2013) proposed a bidirectional exchange. However, a detailed discussion of the exchange processes and their implications on the isotopic source signature is clearly beyond the scope of this paper.

In this study the authors used the same incubation method to determine the stable carbon isotope ratios of CH3Cl emitted from tropical plants (Saito et al., 2008; Saito & Yokouchi, 2008). Moreover the site and most of the sampled plant species were the same in Saito et al. (2013) and in (Saito et al., 2008). Hence, regardless of the nature of the exchange process, we can

safely assume that any potential effect of the bidirectional exchange reported in 2013 is already included in the isotopic source signatures reported in 2008 and that there is no need for a correction of these values as stated in our manuscript.

*8,25-30: "The lifetimes where then forced to ... " is unclear and, if taken literally, questionable. In order to "force the lifetime" to reproduce the seasonal variations at Mace Head assumptions about the seasonality of emission rates are needed. Apart from the very limited a priori information on known sources, this would create circular reasoning about the seasonality of the "unknown source".*

**Authors reply:** We apologize for the confusion. More precisely the loss rates were forced to reproduce the seasonal variations at Mace Head. This forcing accounts for the spatio-temporal variabilities in the OH and $CH_3Cl$ concentration fields, which cannot be adequately represented in a two box model.
Several 2- and 3-D model studies (Lee Taylor et al., 2001; Yoshida et al., 2006; Xiao et al., 2010) provide a quite consistent picture of the seasonal and spatial variations in the overall emissions that serves as a base for this forcing. It should be noted that the seasonality of the unknown source is not discussed in our manuscript. Further, as outlined before tropospheric $\delta^{13}C(CH_3Cl)$ against seasonal and random variations in the isotopic source signatures.

*9,11: spore space*
**Authors reply:** This should be pore space and has been corrected.

*9,31: "nicely resembles"?*
**Authors reply:** We removed "nicely"

*10,11: This conclusion depends strongly on assumption about the seasonal variation of the isotopic signature of emissions.*

**Authors reply**: This conclusion is based on yearly averaged source signatures. As outlined in our response above seasonal variations of the isotopic source signatures have only a minor effect on seasonal variations in the tropospheric $\delta^{13}C(CH_3Cl)$ because of the strong dilution effect of the tropospheric background. A more detailed discussion regarding this issue has been added to the supplementary information.

*References should be checked carefully, I noticed that some cited publications are not included in the references.*

**Authors reply:** We thoroughly checked all references included in the revised manuscript.

*Table 4: The meaning of range should be explained (a ± n\* σ probability range, a percentile, a firm limit based on a given probability?)*

**Authors reply:** Please refer to response of comment above.

---

## Author Comment (AC4) · 6 Jan 2019

We would like to thank J. Laube for this comment that inspired some further thoughts on the missing $CH_3Cl$ source

Increased anthropogenic emissions (Li et al., 2017) may to some extent help to close the $CH_3Cl$ budget. From a purely isotopic perspective, they would fit into the picture and in conjunction with the Central East Asian outflow (Oram et al., 2017), they might appear as tropical emissions. But, this further requires a careful evaluation of increased anthropogenic emissions with respect to the interhemispheric distribution of the $CH_3Cl$ sources. The lack of a substantial interhemispheric gradient in annual mean mixing ratios (Rassmussen, 1980; Simmonds et al. 2004) along with a small interhemispheric gradient in the annual mean OH mixing ratio implies a quite even interhemispheric distribution of the $CH_3Cl$ sources. In line with previous model studies (Yoshida et al, 2006; Xiao et al., 2010 we assumed NH:SH ratio of 1.25 in our model. A substantial increase of the anthropogenic emissions, mainly showing up in the northern hemisphere, would require an adjustment of the interhemispheric ratio of the other sources and or the sinks.

There are other sources and formation processes, being worth to be (re)-evaluated with respect to the missing $CH_3Cl$ emissions.

In the 2014 WMO Ozone Assessment (Carpenter et al. 2014) the $CH_3Cl$ emissions from biomass burning have been revised downward. On the other hand two recent studies (Santee et al., 2013; Umezawa et al., 2014) have shown that enhanced levels of $CH_3Cl$ in the upper troposphere can often be attributed to biomass burning and highlighted the importance of the $CH_3Cl$ biomass burning source. . Hence a careful revision of the biomass burning source may help to reduce the gap between $CH_3Cl$ sinks and sources.

Keppler et al. (2000) reported on the iron catalyzed formation of $CH_3Cl$ from methoxy phenols of soil organic matter. While organic rich peat lands have long been recognized as a source for $CH_3Cl$ (Dimmer et al., 2001), soils in general have been considered as a net sink for $CH_3Cl$ but with highly variable gross emission and deposition fluxes. A recent study has shown remarkable net emissions of $77 \pm 2.8$ µg m$^{-2}$d$^{-1}$ from organic rich temperate forest soils (Redeker et al. 2012). In addition substantial production of $CH_3Cl$ has also been demonstrated for hypersaline soils (Kotte et al. 2012). Manley et al. (2006) reported average fluxes of 3960 nmol m$^{-2}$d$^{-1}$ from bare saltmarsh soils. Thus on a global scale, soils may probably turn from a net sink to a net source.

Moore (2008) proposed a photochemical production pathway for $CH_3Cl$ in saline waters from methoxy phenols. The reaction comprises the photochemical cleavage of the aromatic ring system followed by a nucleophilic substitution with chloride. This mechanism might explain strongly enhanced $CH_3Cl$ concentrations of up to 2 nmol $L^{-1}$ reported from European estuaries (Christoph et al. 2002). With the vast amounts of DOC exported by the large tropical rivers this source may become significant on a global scale.

Methoxyphenols as well as chloride are widespread in the environment. The photochemical mechanism proposed by Moore (2008) may also be relevant for saline surface soils (see above). Further, phenols and methoxy phenols can contribute to 20–40 % of particulate mass from burning hardwood and softwood (Hawthorne et al., 1989). Given this, the photochemical production pathway proposed by Moore (2008) may also be relevant with respect to biomass burning aerosols. Notably, such a mechanism could also take place for quite some time after the burning event in aged biomass burning plumes and in particular after mixing with sea salt aerosols.

**References**

Carpenter, L. J., Reimann, S., Burkholder, J. B., Clerbaux, C., Hall, B., Hossaini, R., Laube, J., and Yvon-Lewis, S.: Chapter 1: Update on Ozone-Depleting Substances (ODSs) and Other Gases of Interest to the Montreal Protocol, in: Scientific Assessment of Ozone Depletion, Global Ozone Research and Monitoring Project Report,World Meteorological Organization (WMO), 21– 125, 2014.

Christof, O., Seifert, R., and Michaelis, W. (2002): Volatile halogenated organic compounds in European estuaries, Biogeochemistry, 59, 143-160, doi: 10.1023/a:1015592115435

Dimmer, C. H., Simmonds, P. G., Nickless, G., and Bassford, M. R. (2001): Biogenic fluxes of halomethanes from Irish peatland ecosystems, Atmospheric Environment, 35, 321-330, doi: 10.1016/s1352-2310(00)00151-5.

Hawthorne, S. B., Krieger, M. S., Miller, D. J., and Mathiason, M. B.: Collection and quantitation of methoxylated phenol tracers for atmospheric-pollution from residential wood stoves, Environ. Sci. Technol., 23, 470–475, (1989)

Keppler, F., Eiden, R., Niedan, V., Pracht, J., and Schöler, H. F. (2000): Halocarbons produced by natural oxidation processes during degradation of organic matter, Nature, 403, 298-301, doi: 10.1038/35002055.

Kotte, K., Löw, F., Huber, S. G., Krause, T., Mulder, I., and Schöler, H. F.: Organohalogen emissions from saline environments spatial extrapolation using remote sensing as most promising tool, Biogeosciences, 9, 1225-1235, 10.5194/bg-9-1225-2012, 2012.

Li, S., Park, M.-K., Jo, C. O., and Park, S.: Emission estimates of methyl chloride from industrial sources in China based on high frequency atmospheric observations, Journal of Atmospheric Chemistry, 74, 227-243, 10.1007/s10874-016-9354-4, 2017.

Manley, S. L., Wang, N.-Y., Walser, M. L., and Cicerone, R. J.: Coastal salt marshes as global methyl halide sources from determinations of intrinsic production by marsh plants, Global Biogeochemical. Cycles, 20, GB3015, doi: 10.1029/2005gb002578, (2006).

Moore, R. M.: A photochemical source of methyl chloride in saline waters, Environmental Science & Technology , 42, 1933 – 1937, doi: 10.1021/es071920I, (2008).

Oram, D. E., Ashfold, M. J., Laube, J. C., Gooch, L. J., Humphrey, S., Sturges, W. T., Leedham-Elvidge, E., Forster, G. L., Harris, N. R. P., Mead, M. I., Samah, A. A., Phang, S. M., Ou-Yang, C. F., Lin, N. H., Wang, J. L., Baker, A. K., Brenninkmeijer, C. A. M., and Sherry, D.: A growing threat to the ozone layer from short-lived anthropogenic chlorocarbons, Atmos. Chem. Phys., 17, 11929-11941, 10.5194/acp-17-11929-2017, 2017.

Rassmussen, R.A., L. E. Rassmussen, . Khalil, A.K., Dalluge, R . W., Concentration distribution of methyl chloride in the atmosphere. J. Geophys. Res. 85: 7350-7356, 1980.

Santee, M. L., Livesey, N. J., Manney, G. L., Lambert, A., and Read, W. G.: Methyl chloride from the Aura Microwave Limb Sounder: First global climatology and assessment of variability in the upper troposphere and stratosphere, Journal of Geophysical Research: Atmospheres, 118, 13,532-513,560, doi:10.1002/2013JD020235, 2013.

Simmonds, P. G. et al. AGAGE Observations of methyl bromide and methyl chloride at Mace Head, Ireland, and Cape Grim, Tasmania, 1998–2001. Journal of Atmospheric Chemistry 47, 243-269, doi:10.1023/B:JOCH.0000021136.52340.9c (2004).

Yoshida, Y., Wang, Y., Shim, C., Cunnold, D., Blake, D. R., and Dutton, G. S.: Inverse modeling of the global methyl chloride sources, Journal of Geophysical Research: Atmospheres, 111, doi:10.1029/2005JD006696, 2006.

Umezawa, T., Baker, A. K., Oram, D., Sauvage, C., O'Sullivan, D., Rauthe-Schöch, A., Montzka, S. A., Zahn, A., and Brenninkmeijer, C. A. M.: Methyl chloride in the upper troposphere observed by the CARIBIC passenger aircraft observatory: Large-scale distributions and Asian summer monsoon outflow, Journal of Geophysical Research: Atmospheres, 119, 5542-5558, doi:10.1002/2013JD021396, 2014.

Xiao, X., Prinn, R. G., Fraser, P. J., Weiss, R. F., Simmonds, P. G., O'Doherty, S., Miller, B. R., Salameh, P. K., Harth, C. M., Krummel, P. B., Golombek, A., Porter, L. W., Butler, J. H., Elkins, J. W., Dutton, G. S., Hall, B. D., Steele, L. P., Wang, R. H. J., and Cunnold, D. M.: Atmospheric three-dimensional inverse modeling of regional industrial emissions and global oceanic uptake of carbon tetrachloride, Atmos. Chem. Phys., 10, 10421-10434, 10.5194/acp-10-10421-2010, 2010.

---

## Editor Decision (ED1)

Gola et al. (2005) and Sellevåg et al. (2006) have neglected the influence of the reaction of $CH_3Cl$ with $O(^1D)$ in their experiments. It is also missing from their FACSIMILE-based kinetic reaction modelling (supplement to the Gola et al. paper). This reaction is likely to have biased their results for the isotope effect of the $CH_3Cl$ + OH sink. In the present set of experiments, the influence of the $O(^1D)$ reaction is likely to be small (assuming the difference between the measured $CH_4$ + OH isotopic fractionation and the literature value of Saueressig et al. 2001 is due to the presence of $O(^1D)$). See also comments below. I suggest you make a stronger statement on this around 7/24; "unresolved" doesn't really capture the potential experimental error of the Gola et al. experiments.

It would be interesting (and straightforward) to measure the isotope effect of the reaction of $O(^1D)$ with $CH_3Cl$. You do not appear to have done a corresponding control experiment ($O_3$ photolysis without water vapour). Please mention the absence of this control experiment in your discussion.

Use decimal points, not commas, throughout the manuscript (incl. figures and tables).

You sometime refer to control and sometimes to "blank" experiments. Please use the term control experiment throughout the paper.

1/22: For clarity, please write "$^{13}C/^{12}C$ carbon isotope effect ($\varepsilon = k(^{13}C)/k(^{12}C) - 1$)" (or "carbon isotope fractionation").

1/23 & 1/24: Please write either "we found an $\varepsilon$ value of ($x \pm y$) ‰" or "we found $\varepsilon = (x \pm y)$ ‰."

1/27: Replace "fractionation factors" with "values for the carbon isotope effect" or "carbon isotope fractionations".

2/18: Change to "KIEs of the main tropospheric sink reactions ($CH_3Cl$ + OH, $CH_3Cl$ + Cl)".

3/37: Brackets missign around (-26.8±0.2) ‰.

4/1: Measurements against the machine working as do not follow the "identical treatment principle", so the value of $-37.2$ ‰ should be provided for anecdotal purposes only. This is NOT an offset between DI and GC-IRMS method. The offline method represents the calibration of the $CH_3Cl$ standard, which should be used as anchor point to report all subsequent measurements of $CH_3Cl$ by GC-IRMS.

How often was the $CH_3Cl/N_2$ reference standard analysed?

Was the same $CH_3Cl/N_2$ reference standard used in Hamburg and Heidelberg?

4/15 & elsewhere: Please change to "the kinetic isotope effect (KIE, symbol $\varepsilon$)". To avoid ambiguity, whenever you refer to the value of the KIE in the manuscript, please use the quantity symbol $\varepsilon$ rather than the abbreviation KIE, including in the supplement.

4/15 & 19 (3 times): Delete the square brackets around ‰.

4/17: Replace "1000" with "1" (two occasions).

4/18: Replace "enrichment factor" with "kinetic isotope fractionation" or "kinetic isotope effect".

Change to "residual $CH_3Cl$ fraction".

5/4: Negative sign missing before "0.00117".

5/12: Delete "molecule$^{-1}$". There is no such unit.

5/12: Replace "*" multiplication signs with "×".

5/16: Please change to "the partial lifetime of $CH_3Cl$ with respect to OH" (the OH lifetime is much shorter).

5/20: "stable carbon isotope delta values"

5/20 to end of page: Replace "KIEs" with "isotope effects" or "isotopic fractionations" or "$\varepsilon$ = ...", as appropriate.

5/30: Insert "(in terms of absolute magnitude)" after "upper end".

5/37: A 3 % contribution from $O(^1D)$-related loss would have only changed the epsilon value by about 0.3 ‰. This cannot explain the difference to the result reported by Saueressig et al.

In fact, a contribution of 9 % (= (4.7–3.9) / (13–3.9)) from $O(^1D)$-related loss is required to explain the higher (in terms of magnitude) $\varepsilon$ value found here.

7/1: Based on the value of 9 %, the potential contribution of $O(^1D)$-related loss to the overall $CH_3Cl$ loss in your experiment is 2.3 % - still small, but not <1 % as you claim. It is less than for $CH_4$ because the relative reaction rate coefficients favour the reaction with OH over $O(^1D)$ in case of $CH_3Cl$ compared with $CH_4$.

8/15: Komatsu is misspelled.

8/26: Please give the "other" NH and SH source strengths also in Gg $a^{-1}$, for consistency and ease of comparison.

9/19: Without an estimate of the degree of $CH_3Cl$ break-down (e.g. soil mole fraction / atmospheric mole fraction), nothing can be said about the magnitude of the isotope effect based on the soil $\delta(^{13}C)$ value alone. E.g. if only 5 % got broken down, the 2.4 ‰ difference between soil and atmospheric $^{13}C/^{12}C$ ratio would correspond to an kinetic isotope effect of –48.5 ‰.

10/9: Your statement "In this case, increasing the soil sink could even lead to a depletion in the tropospheric δ13C once the apparent KIE of the soil sink becomes smaller than the KIE of the OH sink." does not make sense because the tropospheric δ13C is "fixed" (by observations). Presumably, you want to say that "increasing the soil sink" leads to a decrease in the overall sink isotope effect, which must be balanced by an overall increase (i.e. becoming less negative) in the average source δ13C value.

10/23 & 10/34: The atmospheric measurements impose a strong constraint on the model and should be discussed in their own section (new section number 3.2), immediately following the model description (section 3.1) and before section 3.1.1.

Sections 3.1.1 and 3.1.2 should be renumbered 3.3 and 3.4.

Section 3.2 and 3.3 should be renumbered 3.5 and 3.6.

**Supplement**

Please provide consecutive line numbers for the supplement.

Some of the information in the supplement duplicates the methods section. Please remove anything from the supplement that is already included in the methods section.

- Teflon FEP (not Felon FEP)
- Replace "ppb" with "nmol $mol^{-1}$".
- <1 nmol $mol^{-1}$ $O_3$ (see page 6)
- <500 pmol $mol^{-1}$ $NO_x$ (or <0.5 nmol $mol^{-1}$, which helps avoid using another unit)
- "for at least 4 μmol mol-1 8 h" does not make sense
- "typically cleaned for 6 to 8 h" (delete h after 6)

- What was the make and model of the Teflon fan?
- Teflon is written with uppercase initial T because it is a proper name.
- $O_2$ and $NO_x$ should be written with subscript "2" and "$x$". "$x$" in $NO_x$ should be written in italics.
- Delete "ppm" in "1ppm".
- Should this be "between and 1 and 10 µmol mol$^{-1}$"?
- Delete "~" sign before 25 µmol mol$^{-1}$ and replaced with an actual estimate of the uncertainty of this quantity.
- helium is written with lower-case "h" (or use the chemical symbol "He")
- Valco – capital V

- (also 3/23) Please state the actual actinic flux (in photons per area and time or energy per area and time) as well as the make and model of the solar simulator, in addition to "actinic flux comparable to the sun in mid-summer in Germany".
- Please state the temperature at which the experiments were carried out.
- Delete ")" after 5 %.
- Replace "*" multiplication signs with "×".
- 25000 mmol mol$^{-1}$ should be 25000 µmol mol$^{-1}$. However, even that corresponds to a temperature of 27 ºC. Is this correct?
- Delete "molecule$^{-1}$". There is no such unit.
- Delete "molecules" after $2.9 \times 10^9$ and after $2.0 \times 10^{10}$.

Page 3:

- $H_2$
- Replace "µmole mole-1" with "µmol mol$^{-1}$" (2 times).

- "an unknown source" (not: emissions)
- "were randomly" varied? What do you mean by random? What underlying probability density function did you use? Gaussian? Uniform? Triangular? Beta? Please be more specific.
- How many model runs did you carry out?

Page 6:

- Replace ppb with nmol mol$^{-1}$ and ppt with pmol mol$^{-1}$.

Figure S2: Please add a figure legend to identify the three different colours/symbols (or label the curves as in Fig. S5 below). Use decimal points, not commas. The axis labels should be $y(CH_3Cl)/(\mu mol\ mol^{-1})$ and $y(PFH)/(\mu mol\ mol^{-1})$, to designate that you have plotted mole fractions (the square bracket symbol means "concentration").

The right axis should be labelled "residual fraction $f_t$" (see Eq. 3; you might want to add a cross-reference to this equation in the figure caption). Use the same scale for both mole

fractions (the scaling by 0.5 doesn't seem to be correct anyway based on the information on p. 3/8

The $x$-axis should be labelled with integer numbers (0, 2, 4, ...) to avoid confusion with clock time. The axis label should be "time $t$/h" [this means time divided by hours].

Figure S3: Again, a figure legend is needed and the axis labels should be amended as per the comments on Fig. S2 above. The equation in the caption should be written as $y(CH_3Cl)/(\mu mol\ mol^{-1}) = 133.5\ e^{-0.004t/h}$. Please provide another decimal for the coefficient in the exponent.

Fig. S4: The axis labels should be amended as per the comments on Fig. S2 above.

The equations should be written as

$y(CO_2\ measured)/(\mu mol\ mol^{-1}) = 469.5\ e^{-0.00118t/h}$ and

$y(CH_4\ measured)/(\mu mol\ mol^{-1}) = 3.52\ e^{-0.00117t/h}$

to be dimensionally correct.

Fig. S5: Again, axis labels should be "time $t$/h", "$y(CH_4)/(\mu mol\ mol^{-1})$", "$y(CO_2)/(\mu mol\ mol^{-1})$" and "$y(O_3)/(\mu mol\ mol^{-1})$".

---

## Author Response (AR2)

**Point by point reply to J. Kaiser** (Co-Editor comment are in italics): We would like to thank the Co-Editor for the very constructive evaluation of our manuscript and for the helpful comments to improve the manuscript. In addition to the requested changes some minor changes have been applied and the figures 4s and 5s in the supplement have been reformatted.

*Co-Editor Decision: Publish subject to minor revisions (review by editor) (29 Dec 2018) by Jan Kaiser*
*Comments to the Author (pdf): acp-2018-855-comments-to-author.pdf*
*Comments to the Author:*
*Gola et al. (2005) and Sellevåg et al. (2006) have neglected the influence of the reaction of CH3Cl with O(1D) in their experiments. It is also missing from their FACSIMILE-based kinetic reaction modelling (supplement to the Gola et al. paper). This reaction is likely to have biased their results for the isotope effect of the CH3Cl + OH sink. In the present set of experiments, the influence of the O(1D) reaction is likely to be small (assuming the difference between the measured CH4 + OH isotopic fractionation and the literature value of Saueressig et al. 2001 is due to the presence of O(1D)). See also comments below. You should make a stronger statement (around 7/24; "unresolved" doesn't really capture the potential experimental error of the Gola et al. experiments).*

**Authors response:** Referring to the Co-Editors comment on possible spectral interferenes we added the following sentence on page 7/39 *"*However it appears that the authors have not tested the accuracy of their isotope ratio measurements as a function of the isotopologue mole fraction in the presence of other species with overlapping spectra (HCl, $H_2O$, $O_3$, etc.), e.g. by using a dilution series.*"*

*It would be interesting (and straightforward) to measure the isotope effect of the reaction of O(1D) with CH3Cl. You do not appear to have done a corresponding control experiment (O3 photolysis without water vapour). Please mention the absence of this control experiment in your discussion. Use decimal points, not commas, throughout the manuscript (incl. figures and tables).*

**Authors response:** We indeed did not carry out a control experiment to measure the isotope effect of the reaction with O(1D), but our control experiment in the absence of water vapour but with 2000 $\mu mol\ mol^{-1}$ $H_2$ comes close to such a control experiment. We clarified this on page 5/21:

*"* Potential side reactions with O($^1$D) were not explicitly investigated in our study but because of the reduced OH yield, this experiment allows to constrain potential losses of $CH_3Cl$ due to reaction with (O($^1$D)).*"*

*You sometime refer to control and sometimes to "blank" experiments. Please use the term control experiment throughout the paper.*

**Authors response:** Change applied.

*1/22: For clarity, please write "13C/12C carbon isotope effect (ε = k(13C)/k(12C) − 1)" (or "carbon isotope fractionation").*

**Authors response:** Change applied.

*1/23 & 1/24: Please write either "we found an ε value of (x±y) ‰" or "we found ε = (x±y) ‰."*

**Authors response:** Change applied.

*1/27: Replace "fractionation factors" with "values for the carbon isotope effect" or "carbon isotope fractionations".*

**Authors response:** Change applied.

*2/18: Change to "KIEs of the main tropospheric sink reactions (CH3Cl + OH, CH3Cl + Cl)".*

**Authors response:** Change applied.

*3/37: Brackets missign around (-26.8±0.2) ‰.*

**Authors response:** Change applied.

*4/1: Measurements against the machine working as do not follow the "identical treatment principle", so the value of –37.2 ‰ should be provided for anecdotal purposes only. This is NOT an offset between DI and GC-IRMS method. The offline method represents the calibration of the CH3Cl standard, which should be used as anchor point to report all subsequent measurements of CH3Cl by GC-IRMS.*

**Authors response:** To clarify that this was a calibration, this section has been changed to: "In order to assure compliance with VPDB scale, a single component standard of $CH_3Cl$ (100 µmol mol$^{-1}$ in nitrogen, Linde Germany) was calibrated against a certified $CO_2$ reference standard (Air Liquide, Germany, -26.8± 0.2 ‰ and a solid standard (NIST NBS 18, RM 8543) after offline combustion and analysis via a dual inlet (DI). The results from the DI (n=6) were (-37.2 ±0.1) ‰ for $CH_3Cl$. The respective $\delta^{13}C$ values from the GC-GC/IRMS, measured against the machine working gas (Air Liquide, Germany, ((-26.8± 0.2) ‰) were (-36.1±0.2) ‰ resulting in an offset (DI – 2D-GC-IRMS) of -1.1 ‰ for $CH_3Cl$."

*Was the same CH3Cl/N2 reference standard used in Hamburg and Heidelberg?*

**Authors response:** No the Heidelberg group used a different standard. There may be indeed be a difference in absolute scales as stated on page 4/16.

*4/15 & elsewhere: Please change to "the kinetic isotope effect (KIE, symbol ε)". To avoid ambiguity, whenever you refer to the value of the KIE in the manuscript, please use the quantity symbol ε rather than the abbreviation KIE, including in the supplement.*

**Authors response:** Change applied.

*4/15 & 19 (3 times): Delete the square brackets around ‰.*

**Authors response:** Change applied.

*4/17: Replace "1000" with "1" (two occasions).*

**Authors response:** Change applied.

*4/18: Replace "enrichment factor" with "kinetic isotope fractionation" or "kinetic isotope effect".*

**Authors response:** Change applied.

*Change to "residual CH3Cl fraction".*

**Authors response:** Change applied.

*5/4: Negative sign missing before "0.00117". +6*

**Authors response:** Change applied.

*5/12: Delete "molecule–1". There is no such unit.*

**Authors response:** Change applied.

*5/12: Replace "*" multiplication signs with "×".*

**Authors response:** Change applied.

*5/16: Please change to "the partial lifetime of CH3Cl with respect to OH" (the OH lifetime is much shorter).*

**Authors response:** Change applied.

*5/20: "stable carbon isotope delta values"*

**Authors response:** Change applied.

*5/20 to end of page: Replace "KIEs" with "isotope effects" or "isotopic fractionations" or "ε = …", as appropriate.*

**Authors response:** Change applied.

*5/30: Insert "(in terms of absolute magnitude)" after "upper end".*

**Authors response:** Change applied.

*5/37: A 3 % contribution from O(1D)-related loss would have only changed the epsilon value by about 0.3 ‰. This cannot explain the difference to the result reported by Saueressig et al.*
*In fact, a contribution of 9 % (= (4.7–3.9) / (13–3.9)) from O(1D)-related loss is required to explain the higher (in terms of magnitude) ε value found here.*
*7/1: Based on the value of 9 %, the potential contribution of O(1D)-related loss to the overall CH3Cl loss in your experiment is 2.3 % - still small, but not <1 % as you claim. It is less than for CH4 because the relative reaction rate coefficients favour the reaction with OH over O(1D) in case of CH3Cl compared with CH4.*

**Authors response:** We picked this up and clarified the potential contributions from O(1D)-related losses. On page 5/6 the following sentence was added: "Saueressig et al. (2001) reported an ε value of -3.9 ‰ for the reaction of $CH_4$ with OH. With respect to this value, a contribution of 9% from the reaction with O($^1$D) is required to explain the difference in terms of O($^1$D) loss." and on page we added: "Assuming a contribution of 9% from the reaction with O$^1$D in the $CH_4$ experiment, the reaction with O$^1$D should contribute less than 2.3 % to the observed $CH_3Cl$ loss"

*8/15: Komatsu is misspelled.*

**Authors response:** Change applied.

*8/26: Please give the "other" NH and SH source strengths also in Gg a–1, for consistency and ease of comparison.*

**Authors response:** Change applied.

*9/19: Without an estimate of the degree of CH3Cl break-down (e.g. soil mole fraction / atmospheric mole fraction), nothing can be said about the magnitude of the isotope effect based on the soil δ(13C) value alone. E.g. if only 5 % got broken down, the 2.4 ‰ difference between soil and atmospheric 13C/12C ratio would correspond to an kinetic isotope effect of –48.5 ‰.*

**Authors response:** We agree with this and have deleted the respective statement on page 9/38

*10/9: Your statement "In this case, increasing the soil sink could even lead to a depletion in the tropospheric δ13C once the apparent KIE of the soil sink becomes smaller than the KIE of the OH sink." does not make sense because the tropospheric δ13C is "fixed" (by observations). Presumably, you want to say that "increasing the soil sink" leads to a decrease in the overall sink isotope effect, which must be balanced by an overall increase (i.e. becoming less negative) in the average source δ13C value.*

**Authors response:** We changed the statement to: **"**The isolated effect of the soil sink would result in a maximum enrichment of 3.8‰ in the tropospheric $\delta^{13}$C that reduces to 2.1‰ when accounting for the concurrent reduction in the OH sink. In this case, increasing the soil sink could even lead to a decrease in the overall sink isotope effect once the apparent isotope effect of the soil sink becomes smaller than the isotope effect of the OH sink."

*10/23 & 10/34: The atmospheric measurements impose a strong constraint on the model and should be discussed in their own section (new section number 3.2), immediately following the model description (section 3.1) and before section 3.1.1.*
*Sections 3.1.1 and 3.1.2 should be renumbered 3.3 and 3.4.*
*Section 3.2 and 3.3 should be renumbered 3.5 and 3.6.*

**Authors response:** The sections have not been renumbered so far and we would like the numbering as it is. The sections 3.11. and 3.1.2 are part of the model description and the discussion in section 3.2 refers to this model description. To our opinion the discussion of the atmospheric measurements should be done in the context with the model results However, we are open to further discussions.

*Supplement*
*Please provide consecutive line numbers for the supplement.*

**Authors response:** Change applied.

*Some of the information in the supplement duplicates the methods section. Please remove anything from the supplement that is already included in the methods section.*

**Authors response:** We removed duplicates as far as possible but with respect to the readability of the supplement, some repetitions were necessary.

*Page 1*
*• Teflon FEP (not Felon FEP)*

**Authors response:** Change applied.
*• Replace "ppb" with "nmol mol–1".*

**Authors response:** Change applied.
*• <1 nmol mol–1 O3 (see page 6)*

**Authors response:** Change applied.
*• <500 pmol mol–1 NOx (or <0.5 nmol mol–1, which helps avoid using another unit)*

**Authors response:** Change applied.
*• "for at least 4 µmol mol-1 8 h" does not make sense*

**Authors response:** "4 µmol mol-1" has been removed.

*• "typically cleaned for 6 to 8 h" (delete h after 6)*

**Authors response:** Change applied.

*• What was the make and model of the Teflon fan?*

**Authors response:** The Teflon fan was made in-house from PTFE

*• Teflon is written with uppercase initial T because it is a proper name.*

**Authors response:** Change applied.

*• O2 and NOx should be written with subscript "2" and "x". "x" in NOx should be written in italics.*

**Authors response:** Change applied.

*• Delete "ppm" in "1ppm".*

**Authors response:** Change applied.

*• Should this be "between and 1 and 10 µmol mol–1"?*

*• Delete "~" sign before 25 µmol mol–1 and replaced with an actual estimate of the uncertainty of*

*this quantity.*

**Authors response:** Change applied.

*• helium is written with lower-case "h" (or use the chemical symbol "He")*

**Authors response:** Change applied.

*• Valco – capital V*

**Authors response:** Change applied.

*Page 2*

*• (also 3/23) Please state the actual actinic flux (in photons per area and time or energy per area and time) as well as the make and model of the solar simulator, in addition to "actinic flux comparable to the sun in mid-summer in Germany".*

**Authors response:** A figure and appropriate references have been added to the supplement.

*• Please state the temperature at which the experiments were carried out.*

**Authors response:** The information has been added on page 1/21.

*• Delete ")" after 5 %.*

**Authors response:** Change applied.

*• Replace "*" multiplication signs with "×".*

**Authors response:** Change applied.

*• 25000 mmol mol–1 should be 25000 µmol mol–1. However, even that corresponds to a temperature of 27 ºC. Is this correct?*

**Authors response:** This was a miscalculation, we apologize for. The true mixing ratio was 16000 µmol mol$^{-1}$.

*• Delete "molecule–1". There is no such unit.*

**Authors response:** Change applied.

*• Delete "molecules" after 2.9 × 109 and after 2.0 × 1010.*

**Authors response:** Change applied.

*Page 3:*

*• H2*

**Authors response:** Change applied.

*• Replace "µmole mole-1" with "µmol mol–1" (2 times).*

**Authors response:** Change applied.

*Page 4*

*• "an unknown source" (not: emissions)*

**Authors response:** Change applied.

*• "were randomly" varied? What do you mean by random? What underlying probability density*

*function did you use? Gaussian? Uniform? Triangular? Beta? Please be more specific.*

**Authors response:** We used a Gaussian probability density function have specified this.

*• How many model runs did you carry out?*

**Authors response:** In total 100 seasons were modelled with i: no fractionation assigned to the sinks

showing the imprint of the combined source signature on the tropospheric $\delta^{13}C(CH_3Cl)$, ii: an $\varepsilon$ value

of -11.2 ‰ for the tropospheric loss and an $\varepsilon$ value a KIE of -59‰ for the tropospheric loss. We

added this information to the supplement.

*Page 6:*

*• Replace ppb with nmol mol–1 and ppt with pmol mol–1.*

**Authors response:** Change applied.

*Figure S2: Please add a figure legend to identify the three different colours/symbols (or label the curves as in Fig. S5 below). Use decimal points, not commas. The axis labels should be y(CH3Cl)/(µmol mol–1) and y(PFH)/(µmol mol–1), to designate that you have plotted mole fractions (the square bracket symbol means "concentration").*

*The right axis should be labelled "residual fraction ft" (see Eq. 3; you might want to add a cross-reference to this equation in the figure caption). Use the same scale for both mole fractions (the scaling by 0.5 doesn't seem to be correct anyway based on the information on p. 3/8*

**Authors response:** The scaling was indeed not correct. Requested changes were applied.

*The x-axis should be labelled with integer numbers (0, 2, 4, …) to avoid confusion with clock time. The axis label should be "time t/h" [this means time divided by hours].*

**Authors response:** The x-axis has been labelled with integer numbers.

*Figure S3: Again, a figure legend is needed and the axis labels should be amended as per the comments on Fig. S2 above. The equation in the caption should be written as y(CH3Cl)/(µmol mol–1) = 133.5 e–0.004t/h. Please provide another decimal for the coefficient in the exponent.*

**Authors response:** The x-axis has been labelled with integer numbers.

*Fig. S4:* The slope is -0.004t/h. Requested changes were applied.

*The equations should be written as*

*y(CO2 measured)/(µmol mol–1) = 469.5 e–0.00118t/h and*

*y(CH4 measured)/(µmol mol–1) = 3.52 e–0.00117t/h*

*to be dimensionally correct.*

**Authors response:** Change applied.

*Fig. S5: Again, axis labels should be "time t/h", "y(CH4)/(μmol mol–1)", "y(CO2)/(μmol mol–1)" and*

*"y(O3)/(μmol mol–1).*

**Authors response:** Change applied.

---

## Author Response (AR3)

**Major changes made to the revised manuscript**

Section 3 and the corresponding figures have been reorganized and partly rephrased as suggested by the Co-editor.

On page 3/29, we added the photolysis frequency for Cl2 along with a reference.

The kinetic isotope effect for the reaction of CH4 with OH that was already mentioned on page 6/5 has been added to table 3.

Figure 2 has been revised. The uncertaintie sin the lower panel have been adjusted to be consistent with Figure S6 in the supplement.

Figure S4 in the Supplement has also been revised. Here the y-axes have been rescaled.

[revised manuscript text omitted]